



*Obtaining sub-daily new snow density from automated measurements in high mountain regions*

Kay Helfricht[1], Lea Hartl[1], Roland Koch[2], Christoph Marty[3], Marc Olefs[2]

[1]IGF - Institute for Interdisciplinary Mountain Research, Austrian Academy of Sciences, Innsbruck, 6020, Austria
[2]ZAMG - Zentralanstalt für Meteorologie und Geodynamik, Climate research department, Vienna, Austria
[3]WSL Institute for Snow and Avalanche Research SLF, Davos, Switzerland

*Correspondence to*: Kay Helfricht (kay.helfricht@oeaw.ac.at)

**Abstract.** The density of new snow is sometimes monitored by meteorological or hydrological services at daily time intervals, or occasionally measured in local field studies. However, meteorological conditions and thus settling of the freshly

deposited snow rapidly alter the new snow density until measurement. Physically based snow models and now-casting applications make use of hourly weather data to determine the water equivalent of the snowfall and snow depth. In previous studies, a number of empirical parameterizations were developed to approximate the new snow density by meteorological parameters. These parameterizations are largely based on new snow measurements derived from local in-situ measurements. In this study a data set of automated snow measurements at four stations located in the European Alps is analysed for several

winter seasons. Hourly new snow densities are calculated from the height of new snow and the water equivalent of snowfall. Considering the settling of the new snow and the old snow pack, the average hourly new snow density is 68 kgm⁻³ with a standard deviation of 9 kgm⁻³. Seven existing parameterizations for estimating new snow densities were tested against these data, and most calculations overestimate the hourly automated measurements. Two of the tested parameterizations were capable of simulating low new snow densities observed at sheltered inner-alpine stations. The observed variability in new

snow density from the automated measurements could not be described with satisfactory statistical significance by any of the investigated parameterizations, but relationships between new snow density and wet bulb temperature are partly visible in the automated measurements data. Wind speed is a crucial parameter for the inter-station variability of new snow density, with higher new snow density at more windy locations. Whereas snow measurements using ultrasonic devices and snow pillows are appropriate for calculating station mean new snow densities, we recommend instruments with higher accuracy

e.g. optical devices for better investigations of the variability of new snow densities on sub daily intervals.





## 1 Introduction

In mountain regions there is an increasing demand for high-quality analysis, now-casting and short range forecasts of the spatial distribution of snowfall. The amount of new snow is a crucial parameter for estimating avalanche danger (McClung and Schaerer, 1993). Operational services, such as avalanche warning, road maintenance and hydrology, but also

hydropower companies and ski resorts need reliable information on the depth (HN) and the water equivalent (HNW) of snowfall. Therefore the new snow density (NSD) is needed to convert HN into HNW and vice versa. Information on HN is especially relevant for cold and windy conditions, when measuring HNW is a difficult task because conventional rain gauge measurements are prone to large errors (e.g. Goodison et al., 1998). Recent results of the Solid Precipitation Intercomparison Experiment (SPICE; Nitu et al., 2012) reveal that these errors still exist in standard meteorological measurements (e.g.

Buisan et al., 2016; Pan et al., 2016). Thus, reliable HNW input data for e.g. snow cover models are difficult to obtain (Egli et al., 2009). However, most forecast models calculate estimated HN from HNW on subdaily time intervals and thus new snow density is needed in equal temporal resolution (e.g. Lehning et al., 2002; Roebber et al., 2003; Olefs et al., 2013). Additionally, NSD has a considerable effect on the snow bulk density of the total snowpack (e.g. Schöber et al., 2016)

NSD is influenced by many processes affecting the shape and size of the snow crystals, such as crystal growth in the clouds,

the aggregation or dismantling along the way to the surface of the snow cover and subsequently the metamorphism and compaction in the snowpack. A complex relation between temperature and supersaturation of the air by undercooled water generates a variety of crystal forms in clouds, from needles and plates to branched shapes (e.g. Nayaka, 1951). As the crystal starts to fall, its mass can increase by aggregation of snowflakes or riming of supercooled water. On the other hand, the mass and size of the single crystals can be reduced by sublimation or dismantling, especially in windy conditions just before

accumulation. Generally, warmer temperatures and a high supersaturation facilitate the formation of larger crystals, while very small crystal types are documented over a wider range of temperatures. The high volumetric fraction of air between the branches of dendrites causes low snow densities when accumulated on the surface, whereas smaller crystals are packed more densely once accumulated. Once the snow crystals have accumulated at the snow surface, the density of the fresh snow starts to increase depending on weather conditions and compaction caused by overlaying of snow. Relationships between

predominant snow crystal type, riming properties and snowfall density where already reported by Power et al. (1964) from snowstorm observations in Canada. They suggested that variations in NSD are more related to temperatures at cloud level than at the surface.

A common mean NSD used to calculate snow height from water content is 100 $kgm^{-3}$. Many studies analysed NSD values on a daily basis and confirmed this 10:1-rule as applicable for a first estimate (e.g. Roebber et al., 2003; Egli et al., 2009;

Teutsch, 2009). However, NSD span a wide range and values from 10 to 350 $kgm^{-3}$ have been reported from American and European mountain ranges, with mean values between 70 and 110 $kgm^{-3}$ (e.g. Diamond and Lowry, 1954; LaChapelle, 1962; Power et al., 1964; Judson, 1965; McKay et al., 1981; Meister, 1985; Judson and Doesken, 2000; Valt et al., 2014). Most of the NSD data analysed in these studies were observed on the basis of daily HN measurements using a snow board. The density is calculated from the height of new snow measured with a ruler and the corresponding water equivalent is derived

from an external precipitation device or from weighing the new snow either in solid or melted form (Fierz et al., 2009).

However, the readings of the snow boards are not necessarily performed at the end of the snowfall event. Rather, they are carried out at a certain time of day. Settling of the new snow by its weight and destructive metamorphism may reduce HN and hence increase NSD. When conditions clear up after the snowfall, radiation and higher temperatures may speed up the metamorphism. High wind speeds will also accelerate the metamorphism with respect to the surface energy balance, or

40 simply cause a decrease or increase of HN due to drifting snow. Thus, the timing of the snowfall and the conditions after the daily observations have a significant influence on HN and NSD. Nevertheless, direct measurements of NSD on a sub-daily



basis are hard to obtain. It is laborious to plan the field work in time with hours of peak snow fall rates, when critical HN of several centimetres enable adequate measurements. Since the 1960s ultrasonic rangers have become more common for observing snow depth changes automatically even on sub-hourly time scales (e.g. Goodison et al., 1984; Serreze et al., 1999; Lundberg et al., 2010). They have the advantage of high temporal resolution and are a more objective method compared to

subjective manual measurements of snow depth (Ryan et al. 2008). Beside snow depth (HS), the water equivalent of the snowpack (SWE) is observed operationally using weighing devices such as lysimetric snow pillows (e.g. Serreze et al. 2009; Egli et al., 2009; Lundberg et al., 2010; Krajci et al., 2017) and snow scales (e.g. http://www.sommer.at/en/products/snow-ice/snow-scales-ssg). Upward looking GPR (e.g. Heilig et al., 2009), GPS techniques (e.g. Koch et al., 2014; McCreight et al., 2014) and the combination of both (Schmidt et al., 2015) have been applied in scientific studies to monitor the depth,

SWE and liquid water content of the snowpack. However, these techniques are rather expensive or not yet in use for long-term observations by operational services.

In general, manual and automatic measurements of SWE are prone to a high relative uncertainty and require a certain degree of maintenance, which makes them complex and labour-intensive (Smith et al., 2017). Bridging of the weighing devices (pillow, scale) by emerging crust layers in the snow pack may reduce measured HNW (e.g. Serreze et al., 1999) and some

monitoring of the instruments is necessary to avoid this. Due to such constrains, SWE measurement instrumentation is installed at considerably fewer stations compared to HS instruments, and only at sites with easy access for appropriate maintenance.

Several studies have shown that daily measured NSD can be related to meteorological parameters, although with different degrees of determination. In 1952, Gold and Power showed that the crystal type is related to its estimated formation

temperature. Diamond and Lowry (1954) plotted linear relationships between NSD and surface air temperature and between NSD and air temperature at the 700-mb level, with a better correlation coefficient for the latter (0.639). On the basis of data from 7 stations in Switzerland located between 1250 and 1800 m a.s.l., Meister (1985) showed that NSD does not correlate with the amount of new snow (HN). He concluded that air temperature does not accurately determine NSD. Nevertheless, binning the data into temperature classes results in a statistical equation with a coefficient of correlation of 0.85. Further, he

recommended considering wind speed in addition to air temperature, at least for stations higher than 1800m a.s.l. On the basis of data sets from of Schmidt and Gluns (1991) and the US Army Corps of Engineers (1956), Hedstrom and Pomeroy (1998) developed a power function using the air temperature for which they found a coefficient of determination of 0.84 and a standard error of estimate of 9.3 $kgm^{-3}$. Jordan et al. (1999) introduced an algorithm for assigning NSD within the SNTHERM snow cover model. They added wind dependence to the temperature parameterization of Meister (1985). This

achieved a reduction of the error, but a significant scatter remained between observed and parameterized NSD values.

Meister (1985) did not notice any dependence of NSD on altitude. This was confirmed by a study of Simeral et al. (2005), who analysed NSD across an elevational gradient in the mountains of Northwestern Colorado. In a linear regression analysis they identified the 700 mbar level air temperature as the best single predictor variable. Teutsch (2009) also concluded that NSD of 12 hour intervals at valley stations is best correlated to the wet bulb temperature at mountain stations in close

vicinity ($R^2$ = 0.86).  He also showed that a stratiform precipitation event leads to a wider range of the new snow density (24 to 280 $kgm^{-3}$) than a cold front event (densities from 77 to 96 $kgm^{-3}$). Roebber et al. (2003) used surface and radiosonde data for the classification of snowfall density into the three classes heavy (ratio 1:1 to 9:1), average (ratio 9:1 to 15:1), and light (ratio less than 15:1). The subsequent principal component analysis isolates seven factors that influence the snow ratio: solar radiation (month), low- to midlevel temperature, mid- to upper-level temperature, low- to midlevel relative humidity,

midlevel relative humidity, upper-level relative humidity, and external compaction (surface wind speed and liquid equivalent).



However, most of these new snow density data exist neither on a sub-daily basis nor at high spatial resolution. Thus, more regional studies examined the relationship of NSD on meteorological parameters based on data from mountain stations. Judson and Doesken (2000) found that near-surface air temperature and new snow density at mountain stations could explain 52 % of the variance in snow density. Wetzel et al. (2004) presented a similar degree of correlation of new snow density to

5 temperature at three high-elevation sites. Particularly for high-SWE events, Alcott and Steenburg (2009) showed that NSD is correlated with near-crest-level temperature and wind speed. Lehning et al. (2002) built an empirical calculation of new snow densities for a time interval of 30 to 60 minutes. They used air temperature, surface temperature, relative humidity and wind speed for the regression analysis and achieved an approximate multiple coefficient of determination of 0.83. Schmucki et al. (2014) used an empirical power relation, including air temperature, wind and relative humidity, to calculate the new

snow density for three contrasting sites in Switzerland using SNOWPACK simulations. NSD in short time intervals of one to two hours were analysed by Ishizaka et al. (2015). They measured even lower densities in comparison to NSD estimates using the SNOWPACK density model, especially for aggregate snow crystal types. On the basis of data from Col de Porte (1325m altitude, French Alps), Pahaut et al. (1976) developed a statistical relationship including the melting point of water, air temperature and wind speed. This parameterization is used to calculate the density of new snow in the snow cover model

CROCUS (Vionnet et al., 2012). Wright et al. (2016) presented a statistical analysis of data from 42 seasons of manual daily snow density measurements along with air temperature and wind speed to derive parameterizations to estimate new snow density. However, they end up with a low coefficient of determination in comparison to former studies.

Settling processes within the new snow due to metamorphism and accumulation have to be considered when computing new snow density. Metamorphism takes into account processes at the snow surface such as the dismantling caused by saltation

and collision of the snow crystals under windy conditions. This results in higher snow densities and a more compacted snow layer. Subsequently the overburden mass of the ongoing accumulation densifies the lower snow layers (e.g. Anderson 1976). The most important factors for the settling are the amount of new snow, the grain size, and the temperature of the snow layer (e.g. Steinkogler, 2009; Vionnet et al., 2012). The relative contribution of settling to depth changes is highest in the first hours after a snowfall, but of course increases in total values with increasing time intervals. Thus, differences in NSD

between daily observations and measurements at or directly after the snowfall can be expected with increasing time lapse between snow deposition and observation. Additionally, wind drift after the snowfall and energy input to the snow surface by radiation may alter NSD in comparison to NSD at the time of snowfall. To minimize the possible errors due to diurnal settling effects, e.g. Teutsch (2009) used only snow fall events between UTC 6 pm and 6 am. However, this restriction was again determined by the temporally fixed daily snow density readings.

Whereas several studies have analysed sub-daily, manual NSD measurements, to our knowledge no extensive analysis of automated NSD measurements in hourly intervals over several winter seasons exists. The aim of this study is to assess the value of automated measurements of hourly HN and HNW for the calculation of NSD at different stations and in hourly time interval. Therefore we examine the following questions:

(1) Are automated measurements of HN and HNW suitable for the calculation of NSD at hourly interval?

(2) How do the mean and the variability of observed NSD differ between distinct study sites?

(3) How well do established density parametrisations represent observed hourly NSD values?

To this end, we calculated NSD from hourly snow depth changes (HN) and hourly SWE changes (HNW) on the basis of data from ultrasonic rangers and snow pillows, respectively. The mean values and the variability of hourly NSD are discussed for



observations at four different meteorological stations and compared to calculations using established NSD parameterisations. A critical assessment with outlook on next generation measurements is given in the discussion.

## 2 Data and Methods

Data from four (4) automatic weather stations (AWS) were used in this study. A prerequisite for the station selection was the
combined measurement of HS and SWE at each station in addition to the standard meteorological measurements of at least air temperature, relative humidity, precipitation, wind speed and global radiation. Based on this criteria, we analysed the data of two snow stations in Austria, one in Germany and one in Switzerland (Fig. 1, Tab. 1). HS and SWE are measured using ultrasonic rangers and snow pillows, respectively. HS data is measured with a resolution of 1 mm, SWE with a resolution of 0.1 mm. For details regarding the instruments at and the exact location of each AWS, as well as the start and end dates of the
available data coverage, see Table 1.

The Kühroint station (12°57'35.5" E, 47°34'12.4" N, 1420 m a.s.l., Germany) is operated by the Bavarian Avalanche Warning Service. It is a well-equipped and maintained station for snow climate at the northern fringe of the Eastern Alps. It is located in a meadow below treeline.

The Kühtai station (11°00'21.6" E, 47°12'25.6" N, 1970 m a.s.l., Austria) is operated by the Tiroler Wasserkraft AG
(TIWAG) and has a time series of both HS and SWE data since 1987 (Krajci et al. 2017). It is located south of the Inntal valley, but north of the Alpine main ridge, and it is situated in the vicinity of a hydroelectric power plant at the foot of a slope just below the treeline in a wind-sheltered area.

The station at Wattener Lizum (11°38'18.6" E, 47°10'05.5" N, 1994 m a.s.l., Austria) is operated by the Austrian Research Centre for Forests (BFW) of the Federal Ministry of Agriculture, Forestry, Environment and Water Management. This
station is situated in a south-north oriented high alpine valley above the treeline next to the Alpine main ridge.

The station at Weissfluhjoch (9°48'35.7" E, 46°49'46.4" N, 2540 m a.s.l., Switzerland) is operated by the Institute for Snow and Avalanche Research (SLF), which is part of the Swiss Federal Institute for Forest, Snow and Landscape Research (WSL). Weissfluhjoch is the highest station considered in this study. It has a long research history and its data yields the basis for numerous scientific studies. Due to its comparatively windy location on an open flat field in alpine terrain, wind
drift can influence the snow accumulation. Detailed information about the Weissfluhjoch data set can be found in WSL Institute for Snow and Avalanche Research SLF (2015) and in Marty and Meister (2012).

On the basis of coinciding data availability we consider four time periods: 1 October 2013 – 20 May 2015 (data available for all stations, referred to as time period 1), 1 October 2011 – 30 September 2013 (data available for all stations except Weissfluhjoch, time period 2), and 1 January 1987 – 30 September 1999 and 1 October 1999 – 30 September 2011 (data
available for Kühtai only, time periods 3 and 4). The latter separation of the Kühtai data series was chosen due to i) the availability of station wind data and ii) the equal period length of 12 years each.

Data outputs of the AWS are logged at time intervals ranging from 2 to 30 minutes. For comparability between stations, hourly average values were computed for each data set. For global radiation, relative humidity, air temperature and wind speed, the hourly value is the mean of the previous hour. For precipitation it is the sum of the previous hour. To account for
noise in the ultrasonic signal, HS was smoothed using a centred moving average over 3 values in the original data resolution of the respective stations. SWE values were smoothed with the same range to guarantee a similar data handling. The hourly values for HS and SWE are the instantaneous values on the full hour, using the corrected and smoothed time series. For the





snow pillow data the instantaneous hourly SWE value is the value at the full hour. Daily mean values for all parameters were also computed, using an analogous approach. Unless stated otherwise, the hourly values form the basis of all further analysis.

The thermodynamic wet bulb temperature was computed using the psychrometric equation (Sonntag, 1990) and an exact iterative approach. Details on the exact iteration can be found in Olefs et al. (2010). In the following, "wet bulb temperature"

always refers to the thermodynamic wet bulb temperature. The wet bulb temperature depends on relative humidity and air temperature, as well as to a lesser extent on air pressure (see Olefs et al., 2010 for more details and a sensitivity study). A standard barometric equation was used to determine a constant value for air pressure based on the elevation of each station and these constant values were subsequently used in the calculation of the wet bulb temperature.

A necessary condition for all further analysis of the time series was the presence of a precipitation signal at the heated

precipitation gauges in combination with positive snow depth changes. Following the extraction of the data points for which this precondition was fulfilled, a number of further filtering procedures were carried out in order to exclude potentially unsuitable data. Threshold values were set for maximum allowed wet bulb temperature ($T_w < 0°C$) and wind speed ($u < 5$ m/s). The hourly height of new snow (HN) and the water equivalent of snowfall (HNW) were computed as the change in total snow height and SWE per hour.

For further analysis, constraints have to be made in order to avoid low values of HNW and HN, which are prone to errors due to random and systemic measurement uncertainties in HN and SWE, but ensuring a minimum of approx. 100 samples.

To investigate the influence of different HNW and HN thresholds, a distribution matrix was calculated by varying the thresholds in steps of 0.5 millimetres for HNW and 0.5 centimetres for HN, respectively. To account for settling during ongoing snowfall, a compaction correction was applied. We used the approach detailed in Anderson (1976). Destructive

settling is considered for the layer of new snow (HN) for each time step $t$ where the snow height increases. Settling of HS within the old snow pack is also considered. For each relevant time step $t$, HS is defined as the snow pack of the previous time step $t-1$. The destructive settling per second of the new snow (SD_HN) for each time step is given as

$$SD\_HN_t = -0.000002777 * e (0.04 * T_t) \qquad\qquad \text{for NSD} <= 150 \text{ kgm}^{-3} \qquad\qquad (1a)$$

$$SD\_HN_t = SD\_HN_t * e (0.046 * T_t * (NSD_t - 150)) \quad \text{for NSD} > 150 \text{ kgm}^{-3}, \qquad\qquad (1b)$$

where $T$ is the air temperature. Weight settling within the new snow layer is not taken into account. Destructive settling within the old snow layer (SD_HS) is calculated using the same equation, substituting SD_HS for SD_HN and using the bulk density of the snowpack (BSD) calculated from HS and SWE at $t-1$ instead of $NSD_t$. Weight settling within the old snowpack (SW_HS) is given as:

$$SW\_HS_t = -248.976 * HNW_t / 3600000 * e(0.08 * T_t) * e (-0.021 * BSD_{t-1}) . \qquad\qquad (2)$$

The resulting settling factors of SD and SW are multiplied with HS and HN to adjust HN accordingly.

New snow density (NSD) was obtained from the ratio HN to HNW. Outliers below the 5 % percentile and higher than the 95 % percentile were excluded. The NSD data were grouped by wet bulb temperature and wind speed, using bins of 1°C and 0.5 ms$^{-1}$ respectively. A least squares regression was carried out using both the ungrouped data and the median of the grouped data to quantify possible correlations of NSD with wet bulb temperature and wind speed.



A stepwise linear regression analysis was carried out in an attempt to predict hourly new snow density using the available meteorological parameters (temperature, humidity, wind speed, radiation). The stepwise forwards regression analysis was initiated with a constant model, using the p-value for an F-test in the sum of the squared errors to determine whether to add or subsequently remove a term. For each location and time period, the resulting linear model was used to compute a data set

of corresponding predicted new snow densities. To evaluate the performance of the predicting models, the p-value and the coefficient of determination ($R^2$) were computed.

The NSD were compared to the following parameterizations developed in previous studies. In these parametrisations, NSD is a function of meteorological parameters such as air temperature (T), wind speed (u) and relative humidity (rH). The time

interval for NSD readings of the respective study is given in the brackets.

$$\rho_{HP} = 67.92 + 51.25 * e\hat{\ }(T / 2.59) \qquad \text{(Hedstrom and Pomeroy 1998, event/daily)} \qquad (3)$$

$$\rho_D = 119 + 6.48 * T \qquad \text{(Diamond and Lowry 1954, frequent interval during event)} \qquad (4)$$

$$\rho_{LC} = 50 + 1.7 * (T+15)\hat{\ }1.5 \qquad \text{(LaChapelle 1962, event)} \qquad (5)$$

$$\rho_J = 500 * (1 - 0.951 * e\hat{\ }(-1.4 * (278.15 - (T + 273.15))\hat{\ }(-1.15) - 0.008 * u\hat{\ }1.7) \ [-13 < T >= 2.5°C] \qquad (6)$$

$$\rho_J = 500 * (1 - 0.904 * e\hat{\ }(-0.008 * U\hat{\ }1.7 )) \ [T <= -13°C] \qquad \text{(Jordan et al., 1999, event/daily)}$$

$$\rho_V = 109 + 6 * (T - Tf) + 26 * u * 1/2 \qquad \text{(Vionnet et al., 2012, event/daily)} \qquad (7)$$

$$\rho_S = 10\hat{\ }(3.28 + 0.03 * T - 0.36 - 0.75 * \arcsin(\sqrt{(rH / 100)} + 0.3 * \log_{10}(U) ) \ [T >= -14 °C] \qquad (8)$$
$$\rho_S = 10\hat{\ }(3.28 + 0.03 * T - 0.75 * \arcsin\hat{\ }(\sqrt{(rH / 100)} ) + 0.3 * \log_{10}(U) ) \qquad \text{[for: T < -14°C]}$$
$$\text{(Schmucki et al., 2014, event/hourly )}$$

$$\rho_L = 70 + 6.5 * T + 7.5 * T_s + 0.26 * rH + 13 * u - 4.5 * T * T_s - 0.65 * T * U - 0.17 * rH * U + 0.06 * T * T_s * rH \qquad (9)$$
$$\text{(Lehning et al., 2002, event/hourly)}$$

$T_f$ in Eq. 7 refers to the melting point of snow and was approximated as $T_f = 0$ °C (Vionnet et al., 2012). According to Schmucki et al. (2014), we limited the parameter range and set rH to a constant value of 0.8 (80 %) during snowfall and the

25 lower boundary for the wind speed to 2 ms$^{-1}$.

In Eq. 9 $T_s$ is the temperature of the snow surface (Lehning et al., 2002). As this was not available at each station, we used the approximation $T_s = T$. We argue that $T_s$ could not considerably exceed 0°, because a threshold of $T_w$ of 0°C was set (see above). Since only precipitation events are considered, rH can be expected to be high, and thus difference between $T_w$ and T is small.

## 3 Results

Changing the lower threshold values for HN and HNW affects the resulting NSD values considerably. Applying different combinations of thresholds to data from all four stations in period 1 (Tab. 2) yields the distribution shown in Figure 2.





Increasing the thresholds results in a distinct lowering of the number of data remaining for the subsequent analysis. There are certain differences between the stations with respect to high thresholds of HNW. Whereas calculated NSD decrease when applying low HN and high HNW thresholds at Kühtai and Wattener Lizum, NSD increase for equal thresholds at Kühroint and Weissfluhjoch station. At Kühtai and Wattener Lizum station, high HNW values are accompanied by rather high HN. In

contrast, low HN occurring with high HNW at Kühroint and Weissfluhjoch cause high NSD. These two stations are located on the northern fringe of the Alps and the range of temperatures at time of snowfall is higher compared to the range at the other more inner-alpine stations (Fig. 7). This may result from the more significant exposure to precipitation events accompanied by advection of warm air with north to westerly flow conditions. However, these results are based on a small number of values only. In general, the calculated median NSD are rather constant following the 1:1 line of HNW and HN

thresholds (Fig. 2 and 3), again with some exceptions when applying high thresholds of 3 mm / cm, respectively.

Figure 3 shows the performance of the different density parameterizations (Eq. 3 to 9) in comparison to calculated NSD values using equal values for HNW and HN thresholds (i.e. 1:1 line in Fig. 2). At three of the four stations, calculated median NSD is lower than 80 kgm⁻³. Comparatively higher NSD are calculated for Weissfluhjoch station, with values between 85 to 100 kgm⁻³. In general, most of the parametrizations result in higher densities compared to median NSD

computed from measured HNW and HN. At this station, an increase of the parametrisized snow density values using the Equations. 3 to 9 with increasing HN/HNW thresholds is obvious, which may be caused by higher accumulation rates during snowfall events with higher temperatures. However, such an increase cannot be observed in the NSD computed from HNW and HN.

With respect to the results of Figure 2, we decided to use a minimum required hourly increase of 1.5 mm in HNW and 2.0 cm in HN as lower thresholds in order to avoid low values of HNW and HN, but ensuring an appropriate number of approx. 100 samples,. This leads to the exclusion of on average 94 % of all data points that have a precipitation signal and positive snow depth changes (Tab. 2). The conservatively chosen thresholds for HN and HNW cause the strongest reduction in data compared to other meteorological constraints (Fig. 4 and S01 to S08). The wind speed threshold has only a small effect on

the lower stations and is more noticeable at the more windy, higher stations of Wattener Lizum (Fig. S06 and S07) and Weissfluhjoch (Fig. S08). Considering period 1 comprising all stations, the filtering process causes the highest filtering rate at Weissfluhjoch in (2013-2015), with about 6 % of data remaining after applying the thresholds. The highest amount of data reduction is found at Kühtai station, with 5 % of the data remaining after filtering of period 3 and 4. Frequency distributions for HN, HNW, wet bulb temperature ($T_w$) and wind speed (U) of the unfiltered and filtered data are presented for each

station and for each time period in the Figures 4 and S01 to S08.

The NSD values obtained from the filtered data show high variability at all stations and change substantially from one hour to the next (Fig. 5). Nevertheless, NSD values are within a reasonable range. The correction of the HN underestimation caused by settling of the snowpack during snowfall leads to an average reduction of mean NSD of 13.5 % with a standard

deviation ($\sigma$) of 3.7 % or 10.2 kgm⁻³ with a $\sigma$ of 2.6 kgm⁻³, and median NSD of 14.3 % with a $\sigma$ of 5.4 % or 10.5 kgm⁻³ with a $\sigma$ of 3.8 kgm⁻³, respectively (Table 2).

The histograms of NSD (Fig. 5 and S09 to S16) show one-tailed distributions towards higher NSD. Median NSD of the different stations and for different periods range between 66 and 86 kgm⁻³ for uncorrected values and between 54 and 83 kgm⁻³ for NSD corrected for settling (Tab. 2). The compaction correction causes noticeably less change in NSD at

Weissfluhjoch in period 1 (5 % reduction of mean NSD) than at the other time periods and stations. The next closest is Kühroint, also in period 1, with a reduction in NSD of 7 %. Unless otherwise stated in the text, NSD always refers to the corrected densities hereafter.





The regression analysis showed that the short term variability of NSD cannot be explained with corresponding changes in wet bulb temperature, wind speed or any of the other available meteorological parameters in a satisfactory manner. Computing NSD using linear regression models developed for each station from the meteorological data yielded only low correlation with the observed values. The results for linear regressions using wet bulb temperature ($T_w$) and wind (u) as single regressors are shown in Fig. 6 and S17 to S24. An increase of NSD with increasing wet bulb temperature can be identified in a statistically vague manner. The slopes of the least squares regression lines show an increase of NSD with an increase of wet bulb temperature for all stations (Table 3, Figure 6). No consistent relationship between NSD and wind speed could be found neither for single stations nor for different periods at one station. The coefficients of determination ($r^2$) and the significance level (p) for the NSD – wet bulb temperature and NSD - wind speed relationship are inconclusive (Tab. 3) but improve somewhat when the mean and median of the data binned by temperature and wind speed, respectively, are applied as explanatory variables (Tab. 4). The binned analysis based on $T_w$ showed a considerable $r^2$ of more than 0.5 on a 0.01 significance level at Kühroint and Kühtai station, with intercepts of 70 to 80 kgm$^{-3}$ and gradients of about 3 to 4 kgm$^{-3}$ per 1°C.

Although the regressions generally show the expected trends, it must be noted that the variability in the NSD values remains largely unexplained, even testing multiple regressions using additional meteorological parameters. Therefore this approach was not pursued further within this study, and we abandon the idea of publishing any new statistical relationship between meteorological parameters and NSD. Instead a comparison to existing parameterisations of NSD was performed. In general, median values of wet bulb temperature ($T_w$) follow the altitudinal gradient between station locations as expected. Higher NSD were calculated for Weissfluhjoch station, where mean $T_w$ during snowfall was lowest and measured wind speeds were highest in comparison with the three other stations for period 1 (Tab. 2, Fig. 7). With respect to wind speeds, Wattener Lizum is second. Lowest wind speeds at Kühtai station occur together with lowest NSD. Considering the median NSD at the four stations, Weissfluhjoch has the highest median NSD by a large margin with 83 kgm$^{-3}$ in period 1 compared to, respectively, 67, 61 and 66 kgm$^{-3}$ at Kühroint, Kühtai and Wattener Lizum. A relationship between NSD and $T_w$ is obvious for Kühtai station between the different periods, with higher NSD for higher $T_w$. Median NSD at Kühtai station varies within a range of 10 kgm$^{-3}$ (15 %) (Tab. 2). The overall mean hourly NSD of all stations and time periods is 68 kgm$^{-3}$ with a $\sigma$ of 9 kgm$^{-3}$.

Considering the various parameterizations, which use temperature and other meteorological parameters to approximate new snow density (Section 2, Eq. 3 to 9), it is evident that the observed variability of NSD is very poorly represented (Tab. 5). This is true for all seven parametrizations used in this study. Most of them overestimate the median of the observed NSD values (Fig. 3, 9 and 10) and correlation between NSD and snow density approximations are low (Tab. 5). However, some parameterizations produce considerably better results than others for median NSD values. The parameterizations of LaChapelle (1962), Diamond and Lowry (1954) and Vionnet et al., (2012) consistently overestimate NSD (Fig. 9 and 10). The parameterization of Hedstrom and Pomeroy (1998) overestimates NSD at Kühroint, Kühtai and Wattener Lizum station (Fig. 9 and 10), but converges with the median NSD at Weissfluhjoch station for period 1 (Fig. 9, Tab. 5). In general, the NSD simulated using the parameterization of Jordan et al. (1999) are closer to calculated NSD, but median NSD are underestimated for Weissfluhjoch station. Median NSD and NSD range at Weissfluhjoch well simulated using the parameterization of Schmucki et al. (2014), which was fitted to original density data from Weissfluhjoch. Whereas the root mean squared error (R) is small applying this parameterization, the lowest R was achieved for Weissfluhjoch station with the parameterization of Diamond and Lowry (1954). The parameterization of Schmucki et al. (2014) overestimates median values of Kühroint, Kühtai and Wattener Lizum station (Fig. 3 and 9, Tab. 5). The parameterizations of Lehning et al. (2002) and Jordan et al. (1999) result in lowest root mean squared errors (R, Tab. 5) compared to the calculated NSD at Kühroint,





Kühtai and Wattener Lizum station, with slightly lower density values using the parameterization of Lehning et al. (2002) fitting best to the low median NSD values of the Kühtai station.

Thus, the parameterization of Lehning et al. (2002) appears to be the first choice regarding the calculation of hourly new
5  snow densities for high elevated and inner alpine regions. This parameterization requires multiple input parameters. Where such data is not available, the parameterization of Jordan et al. (1999), requiring temperature and wind data only, might be a good alternative. Even though, correlations are low in general, some of the highest Pearson correlation values (r, Tab. 5) were achieved by applying the simpler, linear equations by Diamond and Lowry (1954), LaChapelle (1962) and Vionnet et al. (2012). Basically, this shows once again the fundamental relation between snow density and air temperature.

## 4 Discussion

Hourly NSD values were calculated on the basis of HS and SWE changes measured with ultrasonic rangers and snow pillows, respectively. Both measurement techniques are affected by uncertainties, which are discussed in the following.
From a qualitative point of view, HS measurements using ultrasonic rangers are prone to errors in the assumption of the
signal velocity. Meteorological conditions such as temperature, wind, and humidity influence the signal propagation velocity through the air between the device and the snow surface. A standard correction of ultrasound propagation in dependence of air temperature is applied by default. However, temperature is often measured directly at the device using shielded, but unventilated resistor measurements, which may then considerably be affected by radiation changes. In this study, we considered only the subset of data where a precipitation signal is evident and assume a reduced influence of the mentioned
error source because of increased air turbulence and reduced solar radiation input due to cloud cover. Another documented error source is signal blocking by e.g. dense snowfall or drifting snow, which causes HS peaks. However, with the applied filtering procedure and the additional cut-off at the 5 % and 95 % levels, no such spikes which would cause very low NSD were obvious in the data.

Another critical point in the analysis was the resolution of the HS data. Because of the proposed uncertainty of the ultrasonic measurements, only data in the rounded cm-resolution were originally stored for operational use. This rounding induced additional errors because snow depth changes in the order of 2 mm determine whether values close to a half centimetre are rounded up or down to the full centimetre. This can cause differences of 0.8 cm when considering snow depth changes. To avoid this issue we used the mm-resolution of HS values. A positive precipitation value is a necessary condition for the
subsequent analysis and data filtering (precipitation yes/no). This introduces a slight time lag due to the fact that the snow first has to be melted within the heated precipitation gauges in order to be measured by the sensor.

A well-known issue with snow pillows are bridging effects, e.g. crusts made of ice crystals from melting conditions with strong bonding, which may support the weight of the new snow so that HNW is underestimated, which in turn results in low
NSD with fixed HN. We cannot exclude such data explicitly. However, all three conditions (precipitation yes, HN > 1.5 cm, HNW > 0.1 mm) have to be fulfilled for including values in the analysis, so that data without or with lagged HN increase were not considered. Additionally, the chosen snow stations are well maintained in case of implausible data due to their overall good accessibility. E.g. trenches are dug out around the base area of the snow pillow at Kühtai station to cut off the measured part of the snowpack from lateral bonding. Nevertheless, the measurement uncertainty remains high with ± 1cm
for HN and 0.1cm for HNW. Considering mean HN (Tab. 2) and HNW values, the uncertainty is ± 25 kgm⁻³ or 37 % of the mean density. This value is lower considering higher HN, but increases to 80 % for the combination of minimum HN and



minimum HNW of 1.6 cm and 0.2 mm respectively. The mean uncertainty range of 50 kgm$^{-3}$ is in the same magnitude as the range between the 25 % and the 75 % percentiles for most of the observed NSD. This may explain the low correlations when applying the statistical regressions to NSD values of the present data set. The uncertainties are assumed to be equalized considering mean and median values only for total time periods and separated for bins of wet bulb temperature and wind
speed.

The thresholds chosen for data filtering are based on a best guess estimate on the basis of a first analysis (Fig 2), aiming to avoid low HN and HNW values but ensuring an appropriate number of sample data. Changing the thresholds will change the results (Fig 2.). However, with respect to measurement uncertainties, thresholds have been chosen for most NSD studies
(e.g. Meister, 1985; Teutsch, 2009). There was a considerable fraction of data with positive HS changes, a precipitation signal and positive wet bulb temperatures (Fig. 4 and S01 to S08). Most of these data seem to be paired with very small HS changes and are eliminated for the final data set. Nevertheless, a high fraction of data within the final interquartile range was also filtered off (Fig. 5, grey points).

In general, we showed that analysing NSD for hourly time intervals yields considerably lower NSD values compared to new snow densities from daily measurements. For the latter, the composition of snowfall density is affected by dismantling due to wind drift, settling during the snowfall and ongoing metamorphism between the snowfall and the time of measurement. In contrast, the presented NSD are closer to the time of the snowfall event and density changes over several hours due to e.g. energy exchanges and wind drift at the uppermost snow layer can be excluded. That lower densities have to be applied at
sub-daily time intervals in comparison to daily new snow densities was pointed out in the parameterization developed by Lehning et al. (2002), which was derived from in-situ measurements of NSD in hourly resolution. Comparatively low NSD values close to 50 kgm$^{-3}$were presented by Ishizaka et al. (2016), with an average NSD of 52 kgm$^{-3}$ for aggregated snowflakes and 55 kgm$^{-3}$ for small hydrometeors. They further found a mean NSD of 72 kgm$^{-3}$ for a second group of smaller crystals and 99.4 kgm$^{-3}$ for graupel type hydrometeors. Meister (1985) measured NSD lower than 100 kgm$^{-3}$ even on a daily
basis analysing data with a HN(24h) of more than 0.1 m. However, the time interval and the total height of new snow will cause considerable contributions of destructive snow metamorphism, which increases daily NSD in comparison to hourly NSD values, which in turn can assumed to be lower.

To account for the contribution of snow settling to HS changes during snowfall, a simple approach according to Anderson et
al. (1976) was applied. This first order approximation of settling considers the destructive settling in addition to the settling by overburden snow mass. However, it was simplified with respect to HS, SWE and snow density of only two layers: the new snow and the total snow pack of the previous time step. This assessment reduced the calculated NSD considerably by on average 14 % in mean and median HN. Based on a 15 year data set of Weissfluhjoch (WSL Institute for Snow and Avalanche Research SLF, doi:10.16904/1.) from 1 September 1999 to 31 December 2015, the contribution of settling
relative to HN was calculated using the multi-layer SNOWPACK model (e.g. Lehning et al., 2002) and the approach from Anderson (1976). Results are presented in Fig. 11. While a median relative contribution of settling to HN by 19 % was calculated with SNOWPACK, the approach of Anderson (1976) resulted in lower values of 5 % in median and 9 % in mean. Thus, the settling considered for the presented data can be assumed to be appropriate. Higher contributions of settling would result in lower NSD with increased HN assuming a fixed HNW.

The influence of wind on snow depth changes is significant at more windystations. Snow can be deposited or eroded, which leads to snow depth changes and/or increase/decrease in SWE of the snowpack. To eliminate such cases, the precipitation signal from gauge measurements was used as an independent condition for data filtering. However, snow grains are





dismantled by snow drift, and thus more packed into the layer of new snow during windy conditions even over the course of only one hour. This wind influence may be the reason for higher NSD at Weissfluhjoch station compared to the lower elevated stations. Within this study, the difference of mean NSD is 17 kgm$^{-3}$ between Weissfluhjoch and Kühtai station for period 1. The Kühtai station shows lowest NSD and can be characterised as a wind sheltered area of preferential snow
deposition.

A first attempt to perform a regression analysis based on the observed NSD values in combination with meteorological parameters was not successful because of the high variability of NSD and the low significance levels. Therefore we constrained this analysis to the relationships of NSD to wet bulb temperature ($T_w$) and wind speed (U). In general, even the
statistical significance of the trends of NSD with $T_w$ is vague. We conclude that the value of the analysed data is given by the mean and median NSD and its variation between different stations and time periods, and the considerably lower NSD values in contrast to NSD calculated from existing parameterizations. Nevertheless, a simple relationship between NSD calculated from automated measurement of HS and SWE and temperature was recognized for distinct periods and stations with a similar coefficient of determination in comparisson to the results of e.g. Judson and Doesken (2000), Wetzel et al. (2004) or
Wright et al. (2016).

Mair et al. (2015) evaluated some of the parameterizations also considered in this study. Using a distinctly larger time window for smoothing their HS data (5-hour-average), they calculated median NSD between 75 and 100 kgm$^{-3}$ using the parameterizations of Jordan et al. (1999) and Hedstrom and Pomeroy (1998), which is close to  the results presented in this
study. They also found that using the parameterization of LaChapelle (1962) results in mean NSD higher than 100 kgm$^{-3}$. In general they concluded, that using a constant NSD of 100 kgm$^{-3}$ caused an overestimation of seasonal precipitation by up to 30 %. Conversely, a mean NSD of 70 kgm$^{-3}$ will result in better SWE estimations. This is in accordance with the resulting average NSD of 68 kgm$^{-3}$ calculated from automated measurements within our study.

The comparison of observed NSD and NSD approximated using parameterizations from literature reveals that it has to be carefully considered which parameterization should be used for which application and environment. The temporal resolution and the exposure of the station to wind are certain to determine mean NSD values. Mechanical dismantling and cracking of snow crystals in the air and subsequently at the snow surface increases snow density (e.g. Sato et al., 2008). We cannot relate density differences to crystal types or size from the presented data set. However, the inter-station variability of NSD with
respect to different wind conditions is obvious particularly between the Weissfluhjoch station and the wind sheltered Kühtai station.

The study shows the potential of collocated measurements of HS and SWE for determining NSD. However, the high uncertainty of the data from ultrasonic rangers appears to mask the true variability of NSD. More precise optical devices are
an alternative for HS measurement, since the signal propagation velocity does not depend on meteorological conditions (e.g. Mair et al,. 2010; Helfricht et al., 2016). A combination of manual daily readings at the snow board and the high temporal resolution of automated measurement of SWE and HS seems to be advantageous. Thus, daily readings can be disaggregated, and HN, HNW and NSD values can be calculated once a day minimizing the influence by settling of old snow. A high fraction of snowfall data have been filtered out because of low HN or HNW values close to the accuracy level of the
measurements. With increasing accuracy by e.g. use of optical devices and scales insensitive to bridging effects, a higher fraction of hourly or even sub-hourly NSD can be calculated.





The relatively low densities presented in this study are relevant for e.g. snow drift modules of spatially distributed snow cover models. Simply spoken: the less dense the snow, the more potentially drifting particles. Higher NSD at more windy stations may be caused by mechanical cracking and dismantling of the snow particles in the air and in the snowdrift layer. In contrast, low NSD may result from aggregation of snowflakes when falling, and a more loose accumulation at the snow surface with increased air volume in between the particles. In general, the observed inter-station variability shows the importance of differing between more windy mountain stations and less windy stations in the valleys by e.g. using parameterizations with a dependence of NSD on wind speed (e.g. Lehning et al., 2002). Many of the NSD parameterizations investigated here are used in a multitude of point and spatially distributed snow, hydrological or land surface models around the world to calculate NSD and thus derive total snow depth, also in operational modes. Therefore, our findings are highly relevant and have direct implications for the use of all these models.

We constrained this study to a comparison of stations with similar HS and SWE measurements using snow pillows, only. However, recent studies present the performance of cosmic ray neutron sensors (e.g. Schattan et al., 2017), and thus, other long-term data series such as e.g. from Col de Porte (Morin et al., 2012) may be investigated with a similar approach in future. We would like to kindly encourage everyone who owns a similar data basis of HS and SWE from well-maintained snow stations to contact the authors for further cooperation.

## 5 Conclusion

The aim of this study was to assess the value of automated measurements of snow depth (HS) and snow water equivalent (SWE) to compute new snow density (NSD) on an hourly time interval. Complementary data sets of HS and SWE measurements using ultrasonic devices and snow pillows from four mountain stations were used to calculate the height of new snow (HN) and the water equivalent of snowfall (HNW). Subsequently, hourly new snow densities (NSD) were calculated from HN and HNW considering potential underestimation of HN by settling of the snowpack.

The snow measurements using ultrasonic devices and snow pillows were approved to be applicable for the calculation of station average hourly NSD values. An average NSD of 68 kgm$^{-3}$ with a standard deviation of 9 kgm$^{-3}$ was calculated considering all stations and time periods, which is considerably lower than the often applied value of 100 kgm$^{-3}$. Seven existing parameterizations for estimating new snow densities were tested, and most calculations overestimate NSD in comparison to the results from the hourly automated measurements. Two of the tested parameterizations were capable of simulating low new snow densities at sheltered inner-alpine stations, with the parameterization of Lehning et al. (2002) giving the best approximation. However, the observed variability in NSD from the automated measurements could not be described with appropriate statistical significance by any of the investigated algorithms. Relationships between NSD and wet bulb temperature are obvious at all stations in a statistically vague manner. The wind speed is a crucial parameter for the inter-station variability of NSD, with higher NSD at more windy locations. Nevertheless, the natural variability of NSD is masked using the combination of ultrasonic ranging and snow pillow data for NSD calculation, because of the accuracy of the sensors and snow depth changes due to settling of the snowpack and wind drift. Thus, we conclude that the value of the analysed data is given by the mean and median NSD and its variation between different stations and time periods. Recent developments in optical distance sensors and weighing devices increase accuracy of such snow measurements and hence decrease the uncertainty of subsequent calculations. We therefore recommend the use of high accuracy sensors for the determination of NSD on sub daily intervals. .





### Data

The processed set of SNOWPACK input data from Weissfluhjoch station is available at: *WSL Institute for Snow and Avalanche Research SLF (2015): WFJ_MOD: Meteorological and snowpack measurements from Weissfluhjoch, Davos, Switzerland; WSL Institute for Snow and Avalanche Research SLF; doi:10.16904/1.*

The Kühtai station data are published and available by Krajči et al. 2017.

Data of Kühroint station request from the Bavarian avalanche service.

Data of Wattener Lizum station are available on request from the Austrian Research Centre for Forests (BFW).

### Contribution

Kay Helfricht is the main investigator of this study. Lea Hartl performed snow density analysis within the pluSnow project. Roland Koch performed initial quality control, provision and setup of project database for all station and meta data. Christoph Marty prepared the data of Weissfluhjoch station, contributed fruitful discussions and helped to focus the analysis and the manuscript. Marc Olefs contributed significantly to analysis and discussions as the main project partner within the framework of pluSnow project.

### Acknowledgements

The pluSnow project is financed by the Gottfried and Vera Weiss Science Foundation (WWW). The project funding is managed in trust by the Austrian Science Fund (FWF): P 28099-N34. Project duration 10/2015 - 09/2018. The authors want to thank the colleagues of the Tiroler Wasserkraft AG (TIWAG), of the Federal Research and Training Centre for Forests (BFW) and of the Bavarian avalanche service for data provision. In particular we are grateful for the close collaboration by Johannes Schöber (TIWAG) and Reinhard Fromm (BFW). We also want to thank Michael Lehning and Charles Fierz for their helpful comments and fruitful discussion of the results.

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





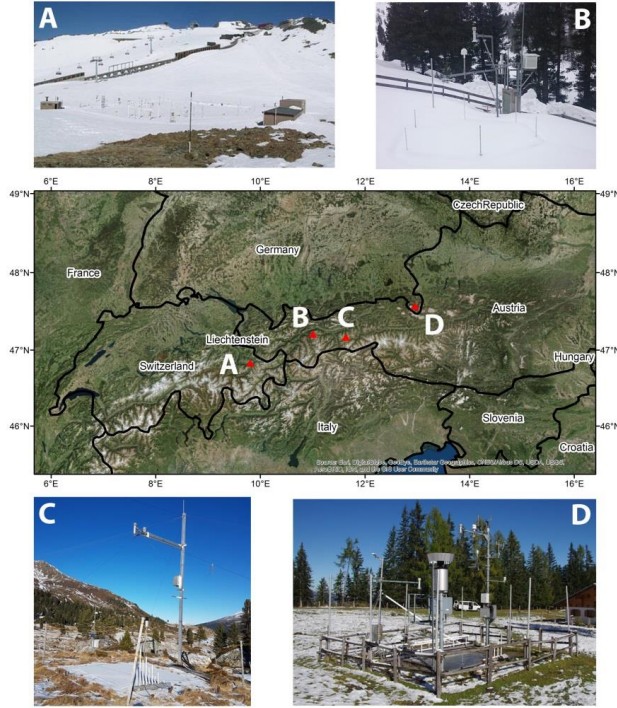

**Figure 1: Map of the station locations. Pictures are given for A) Weissfluhjoch station, B) Kühtai station, C) Wattener Lizum station and D) Kühroint station.**





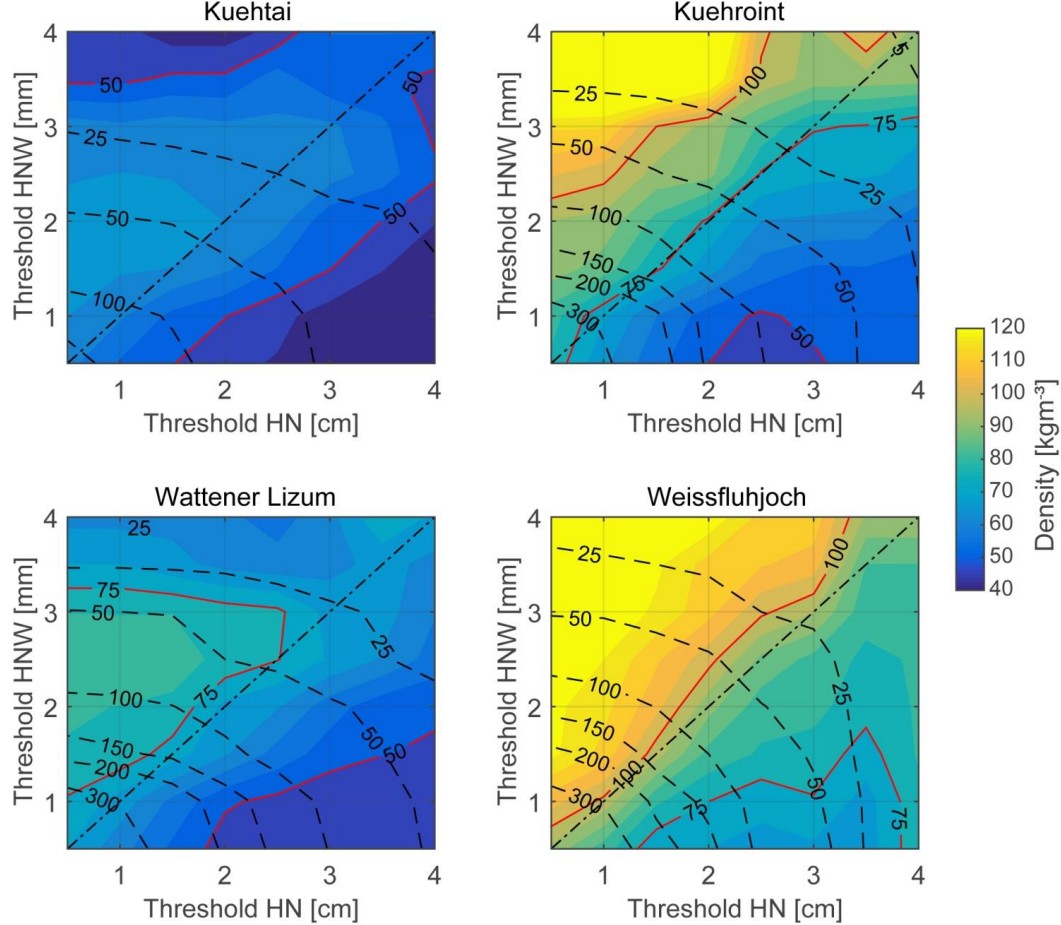

**Figure 2: Median new snow densities (colour scale) calculated using all data exceeding the thresholds of the height of new snow (HN) and the water equivalent of snowfall (HNW) for the period 1 (1 Oct 2013 - 20 May 2015). Note that multiples of 25 kgm$^{-3}$ are highlighted with red contour lines. The black dashed lines give the count of the hourly data remaining for the calculation using the thresholds. The straight dot-dashed lines show equal thresholds of HN and HNW and corresponding results are shown in Fig. 3.**

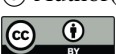



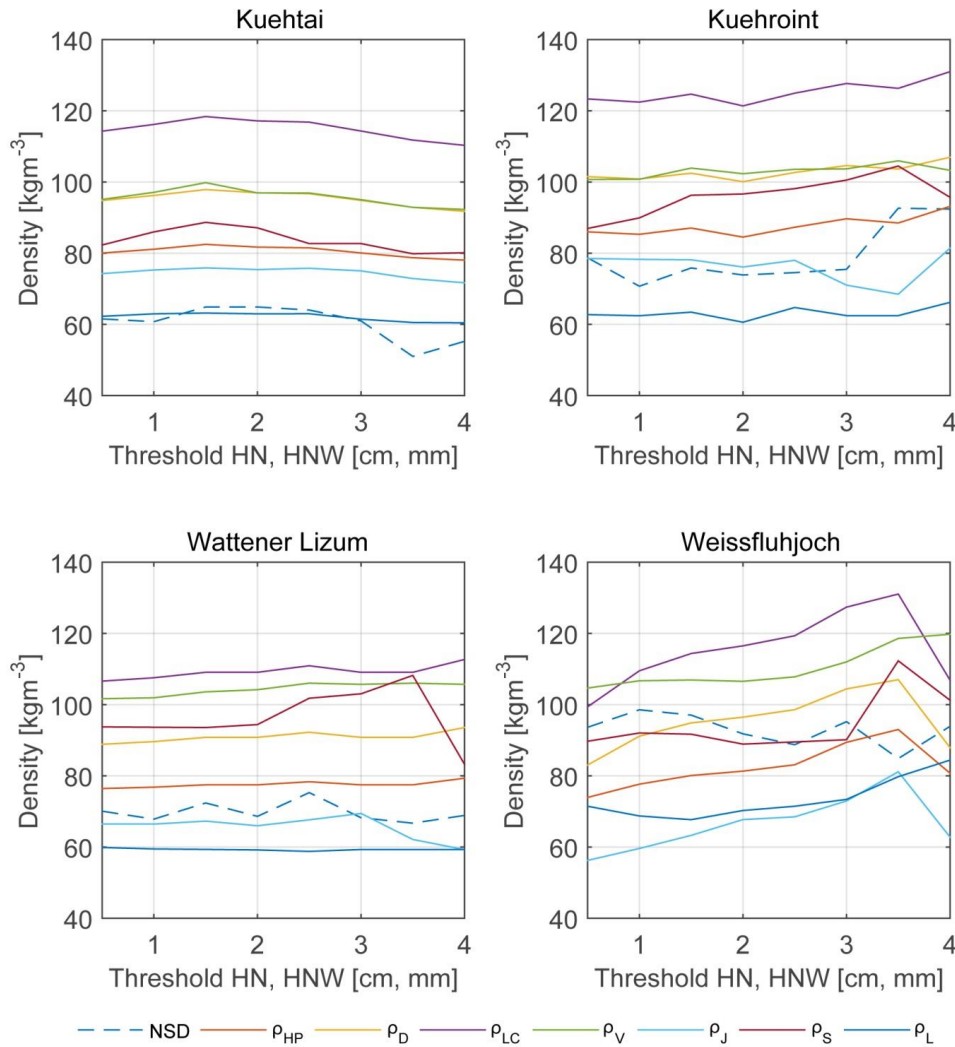

**Figure 3: Median new snow densities calculated using all data exceeding the equal thresholds of the height of new snow (HN) and the water equivalent of snowfall (HNW) for the period 1 (1 Oct 2013 - 20 May 2015). Data of the blue dashed line correspond to the dot-dashed line in Fig. 2. The coloured lines give the results calculated using parameterizations developed in previous studies (see section 2).**





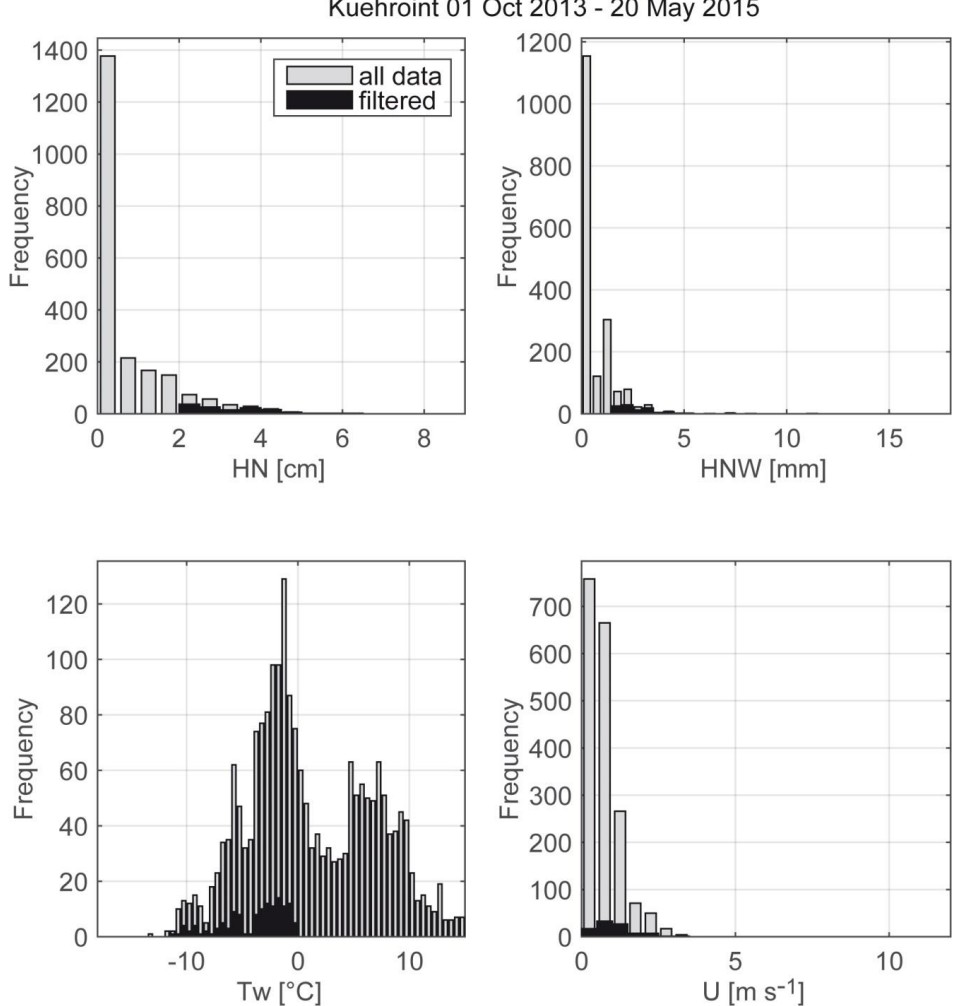

**Figure 4: Histogram plots of all data consisting precipitation signal and positive hourly HS changes ($n_P$, grey) and data filtered with the thresholds HN > 2 cm, HNW > 1.5 mm, Tw < 0° C and U < 5 ms$^{-1}$ ($n_{th, black}$) at Kühroint station for the period 1 (1 Oct 2013 - 20 May 2015). Note that similar figures are available in the supplement (Fig. S01 – S08) for all stations and all time periods considered in this study.**





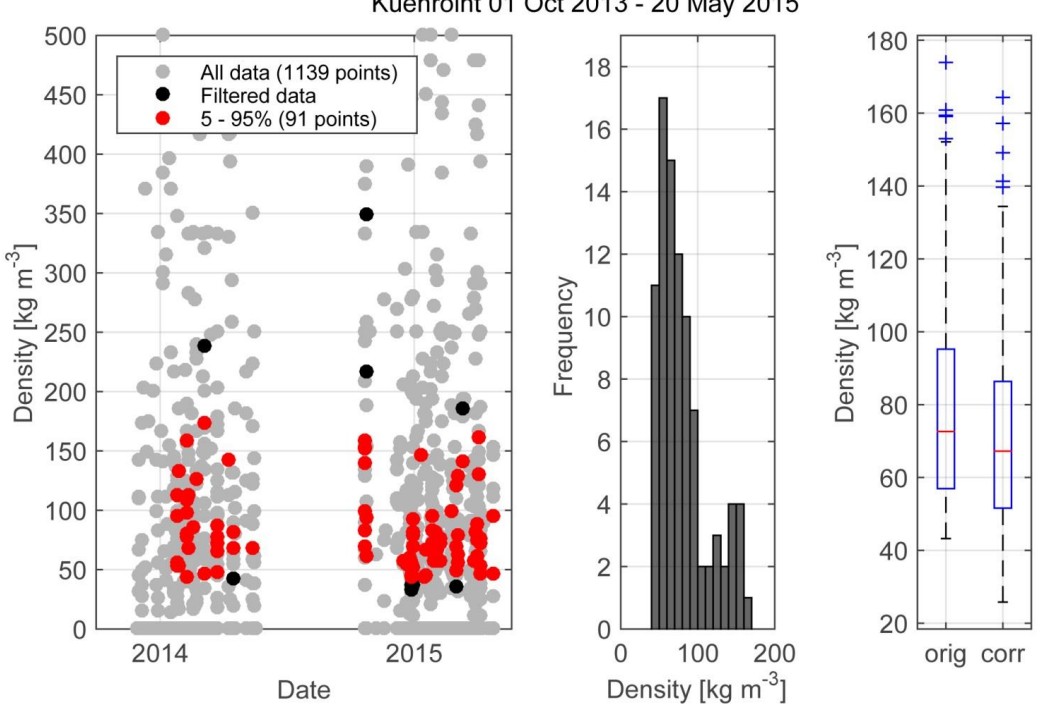

**Figure 5: Distribution of calculated new snow densities at Kühroint station for the period 1 (1 Oct 2013 - 20 May 2015): Left: All data with precipitation signal and positive HS change ($n_P$, grey dots), of all data filtered with thresholds HN > 2 cm, HNW > 1.5 mm, Tw < 0°C and U < 5 ms$^{-1}$ (red and black dots) and filtered data reduced by cutting off at 5 % and 95 % percentiles ($n_{th}$, red dots). Middle: Histogram of all filtered densities ($\rho_{th}$). Right: The boxplot showing median, 25 % and 75 % interquartile range of uncorrected densities and densities corrected for settling of the snowpack. Note that similar figures are available in the supplement (Fig. S09 – S16) for all stations and all time periods considered in this study.**





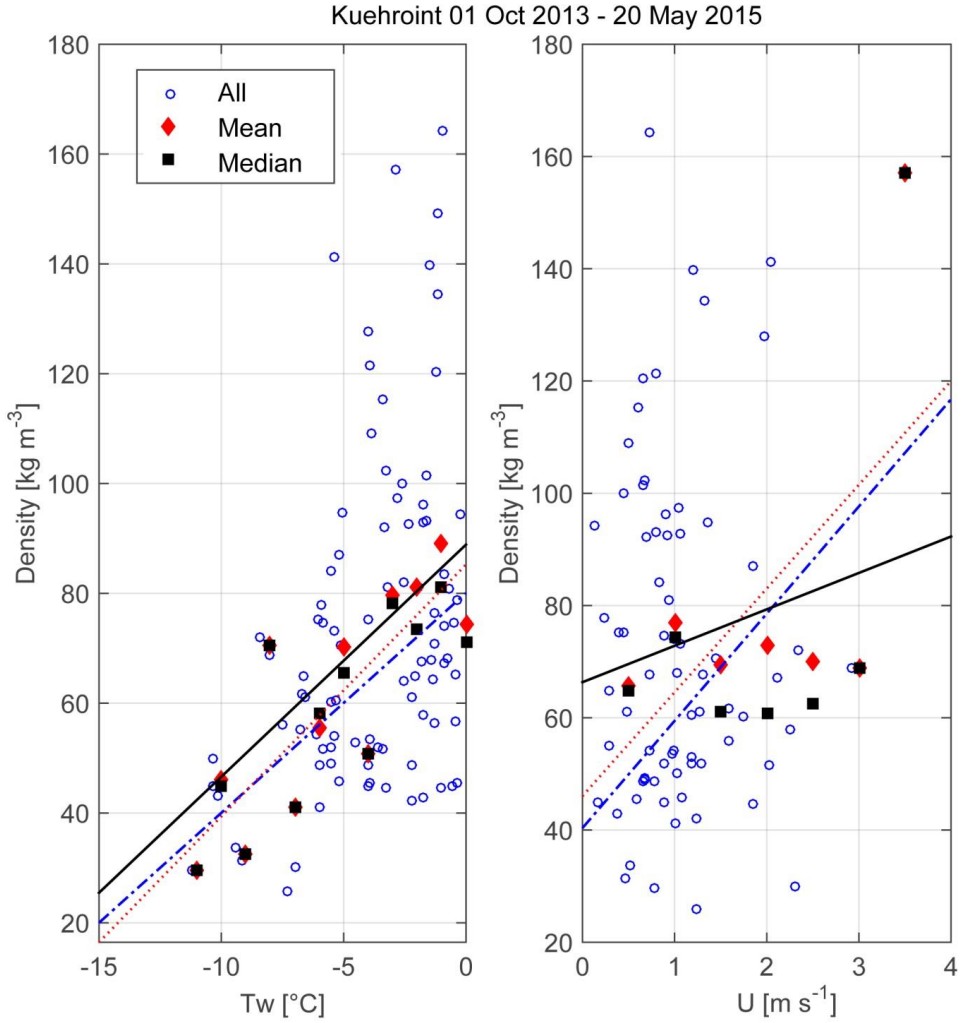

**Figure 6: Linear regression between the corrected densities ($\rho_{corr}$) as dependent variable and wet bulb temperature ($T_w$, left) as well as wind speed (U, right) as explanatory variables for all filtered value pairs ($n_{th}$, blues dots, dashed blue line) at Kühroint station for the period 1 (Oct 2013 - 20 May 2015), and for the class mean (red diamonds, dotted red line) and median (black squares, solid black line) of binned 0.5° K classes and of binned 0.5 ms$^{-1}$ classes, respectively. Corresponding numbers are given in Tab. 3 and 4.**





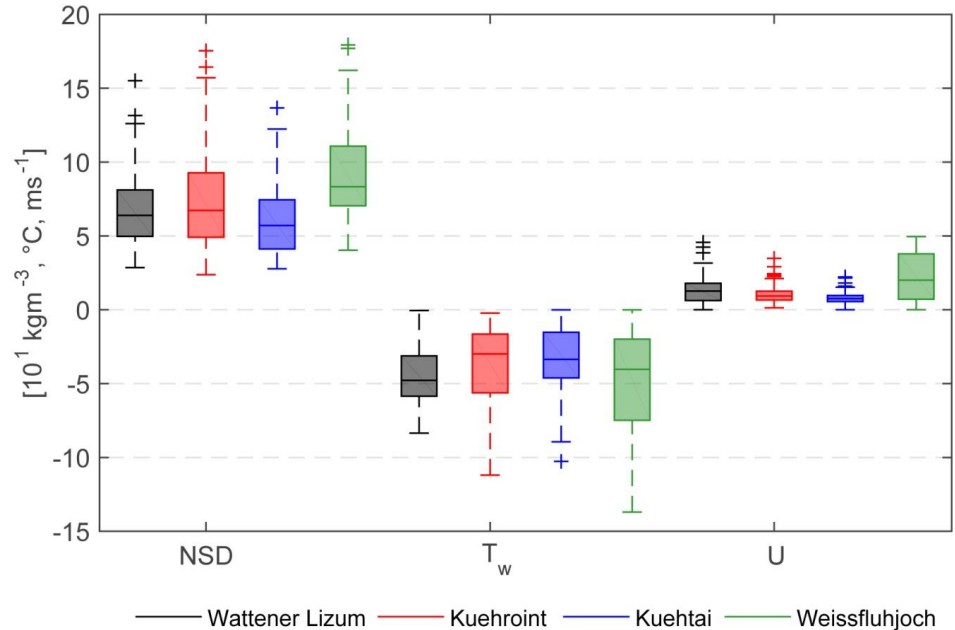

**Figure 7: Boxplot (Median, 25% and 75% percentiles, 1.5 x interquartile ranges, outliers) of calculated new snow densities (NSD) based on observations, wet bulb temperature ($T_w$) and wind speed (u) for filtered snowfall events (Tab. 2) at all four stations within period 1 (1 Oct 2013 - 20 May 2015).**

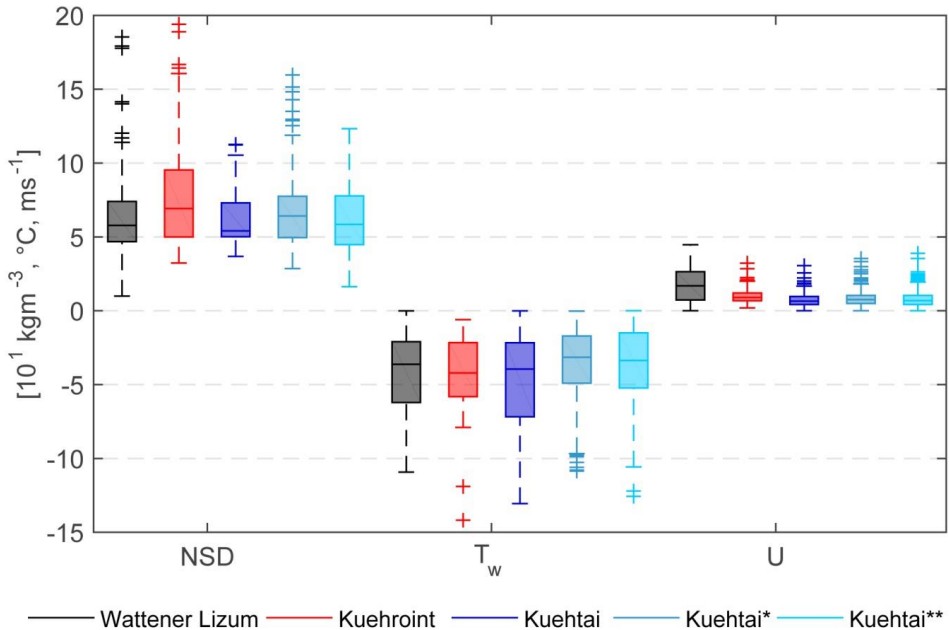

**Figure 8: Boxplots (Median, 25% and 75% percentiles, 1.5 x interquartile ranges, outliers) of calculated new snow densities (NSD) based on observations, wet bulb temperature ($T_w$) and wind speed (u) for filtered snowfall events (Tab. 2) at three stations within period 2 (1 Oct 2011 - 01 Oct 2013) and at Kühtai station within period 3 (index *, 01 Oct 1999 - 30 Sep 2011) and period 4 (index **, 27 Feb 1987 – 30 Sep 1999).**



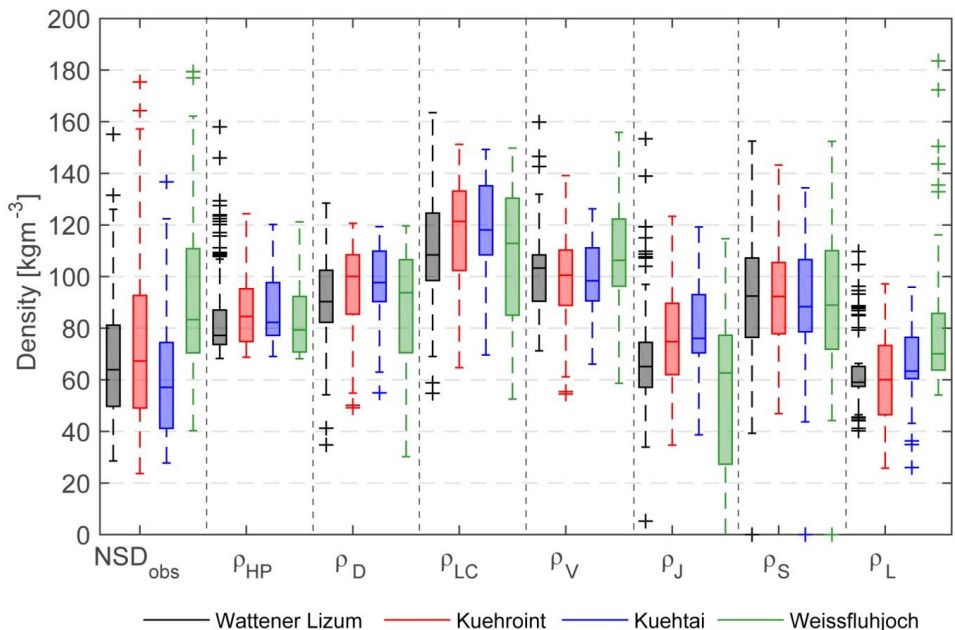

**Figure 9: Boxplots (Median, 25% and 75% percentiles, 1.5 x interquartile ranges, outliers) of calculated new snow densities (NSD) based on observations and densities calculated using parameterizations developed in previous studies (see section 2) all four stations within period 1 (1 Oct 2013 - 20 May 2015).**

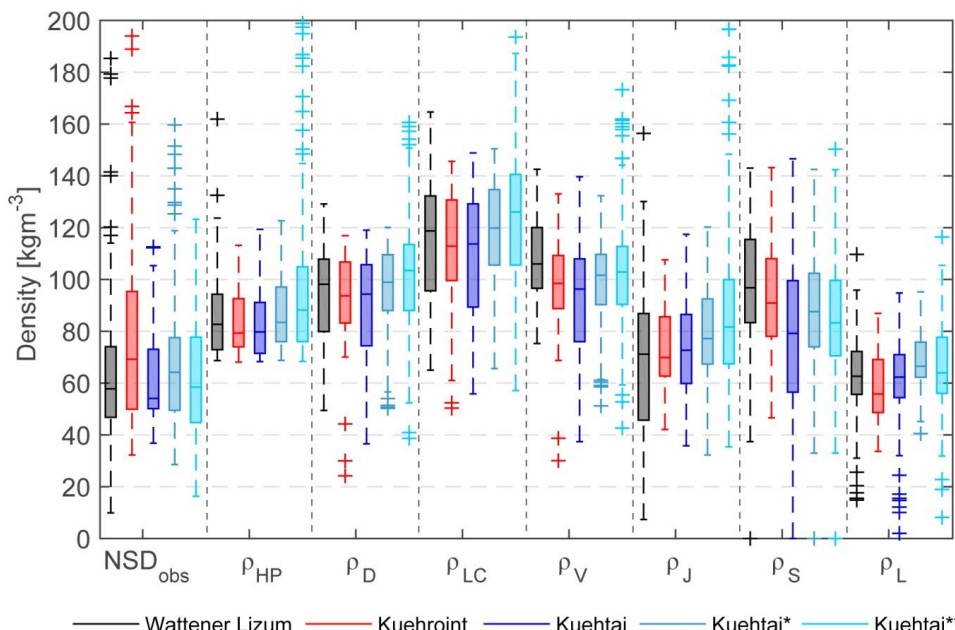

**Figure 10: Boxplots (Median, 25% and 75% percentiles, 1.5 x interquartile ranges, outliers) of calculated new snow densities (NSD) based on observations and densities calculated using parameterizations developed in previous studies (see section 2) at three stations within period 2 (1 Oct 2011 - 01 Oct 2013) and at Kühtai station within period 3 (index *, 01 Oct 1999 - 30 Sep 2011) and period 4 (index **, 27 Feb 1987 – 30 Sep 1999) .**





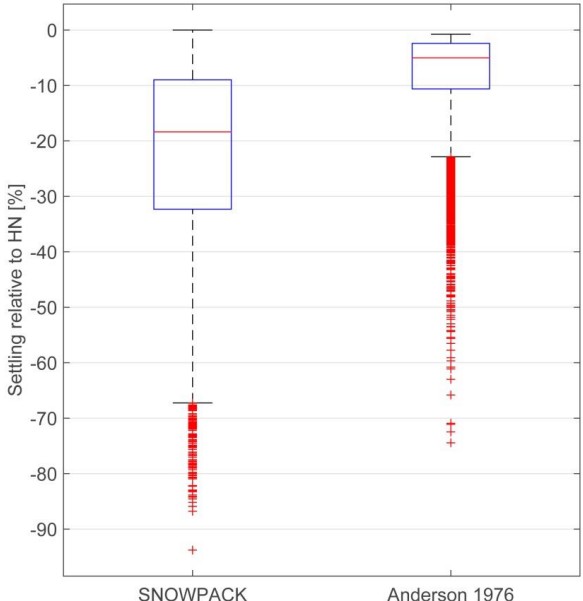

**Figure 11: Boxplots (Median, 25% and 75% percentiles, 1.5 x interquartile ranges, outliers) of settling relative to hourly new snow heights (HN) modelled with SNOWPACK and using the Anderson approach.**




**Table 1: Coordinates and data availability of the four snow stations are given. The instrumentations for measuring snow depth (HS), snow water equivalent (SWE), temperature (T), relative humidity (rH), precipitation (P), wind speed (u) and global radiation (r) are listed.**

| | | Kühroint | Kühtai | Wattener Lizum | Weissfluhjoch |
|---|---|---|---|---|---|
| Station Abbreviation | | KRO | KTA | WAL | WFJ |
| Location | East | 12°57'35.5'' | 11°00'21.6'' | 11°38'18.6'' | 9°48'35.7'' |
| | North | 47°34'12.4'' | 47°12'25.6'' | 47°10'05.5'' | 46°49'46.4'' |
| | z (m a.s.l.) | 1420 | 1970 | 1994 | 2540 |
| Data | | 01 Jan 2011 - 02 Dec 2015 | 27 Feb 1987 - 20 May 2015 | 01 Oct 2010 – 30 Dec 2016 | 01 Oct 2013 - 29 Sep 2015 |
| Instruments | HS | Sommer USH 8 | Sommer USH 8 | Sommer USH 8 | Campbell Scientific SR50A |
| | SWE | Sommer Snow Scale SSG | OTT Thalimedes Shaft Encoder, Endress+Hauser Deltapilot M | Sommer Snowpillow | Sommer Snowpillow |
| | T | Rotronic MP408 | Kroneis NTC | Vaisala HMP45C | Rotronic Hydroclip S3 |
| | rH | Rotronic MP408 | Pernix hair hygrometer | Vaisala HMP45C | Rotronic Hydroclip S3 |
| | P | Sommer NIWA/Med-K505 | Ott Pluvio since 2001, custom built tipping bucket before | Sommer NIWA/Med-K505 | Lambrecht Pluvio 1518 H3 |
| | u | Young 05103 | Kroneis cup anemometer + vane | YOUNG Wind Monitor | Young 05103 |
| | r | Schenk 8101 | Schenk 8101 | Kipp&Zonen CM21 | Kipp&Zonen CM21 |
| Comments | | | data gap winter 2012/13, wind regionalized from 1999 | Meteorological measurements at 2041 m a.s.l. | |





**Table 2: Time periods analysed in this study with mean and median of hourly values for the height of new snow (HN), wet bulb temperature ($T_w$), wind speed (U), calculated densities from observed values (ρ) and calculated densities corrected for settling of the snowpack ($ρ_{corr}$). The results are valid for the data filtered with thresholds HN > 2 cm, HNW > 1.5 mm, Tw < 0° C and U < 5 ms$^{-1}$ ($n_{th}$) values as a subset of all data consisting precipitation signal and positive HS change ($n_P$).**

| Station | # | Period | count data | | HN [cm] | | Tw [°C] | | U [ms$^{-1}$] | | ρ [kgm$^{-3}$] | | $ρ_{corr}$ [kgm$^{-3}$] | |
|---|---|---|---|---|---|---|---|---|---|---|---|---|---|---|
| | | | $n_p$ | $n_{th}$ | mean | median | mean | median | mean | median | mean | median | mean | median |
| KRO | 1 | 1 Oct 2013 - 20 May 2015 | 1139 | 91 | 3.2 | 3.1 | -3.9 | -3.0 | 1.1 | 0.9 | 82 | 73 | 73 | 67 |
| | 2 | 1 Oct 2011 - 30 Sep 2013 | 1576 | 118 | 3.4 | 3.1 | -4.2 | -4.2 | 1.0 | 0.9 | 87 | 77 | 74 | 69 |
| KTA | 1 | 1 Oct 2013 - 20 May 2015 | 579 | 53 | 3.8 | 3.3 | -3.4 | -3.4 | 0.8 | 0.8 | 70 | 69 | 61 | 61 |
| | 2 | 1 Oct 2011 - 30 Sep 2013 | 506 | 36 | 3.3 | 2.8 | -4.8 | -4.0 | 0.8 | 0.7 | 75 | 66 | 60 | 54 |
| | 3 | 1. Oct 1999 - 30 Sep 2011 | 5293 | 252 | 3.5 | 3.2 | -3.5 | -3.2 | 0.8 | 0.8 | 74 | 74 | 64 | 64 |
| | 4 | 27 Feb 1987 - 30 Sep 1999 | 7958 | 387 | 3.7 | 3.3 | -3.6 | -3.4 | 0.8 | 0.7 | 74 | 75 | 61 | 59 |
| WAL | 1 | 1 Oct 2013 - 20 May 2015 | 1248 | 111 | 3.6 | 3.4 | -4.3 | -4.8 | 1.3 | 1.3 | 76 | 72 | 68 | 66 |
| | 2 | 1 Oct 2011 - 30 Sep 2013 | 1588 | 126 | 3.9 | 3.5 | -4.3 | -3.6 | 1.7 | 1.7 | 71 | 69 | 62 | 58 |
| WFJ | 1 | 1 Oct 2013 - 20 May 2015 | 1619 | 100 | 3.0 | 2.7 | -4.9 | -4.0 | 2.2 | 2.0 | 95 | 86 | 91 | 83 |





**Table 3: Results of a single linear regression between the corrected densities ($\rho_{corr}$) as dependent variable and wet bulb temperature ($T_w$) as well as wind speed (U) as explanatory variables for all filtered data points ($n_{th}$). The corresponding coefficient of determination ($r^2$) and the significance level (p) are presented.**

| Station | Period # | $T_w$ | | | | U | | | |
|---|---|---|---|---|---|---|---|---|---|
| | | Intercept | $d\rho_{corr}/dT_w$ | $r^2$ | p | Intercept | $d\rho_{corr}/dT_w$ | $r^2$ | p |
| KRO | 1 | 88.87 | 4.23 | 0.15 | 0.00 | 66.34 | 6.49 | 0.02 | 0.26 |
| | 2 | 86.22 | 2.75 | 0.05 | 0.01 | 74.09 | -1.88 | 0.00 | 0.74 |
| KTA | 1 | 73.88 | 4.53 | 0.14 | 0.00 | 58.10 | 3.59 | 0.00 | 0.69 |
| | 2 | 66.76 | 1.56 | 0.04 | 0.22 | 55.63 | 5.94 | 0.04 | 0.22 |
| | 3 | 70.34 | 1.72 | 0.06 | 0.00 | 66.43 | -2.55 | 0.01 | 0.18 |
| | 4 | 70.63 | 2.59 | 0.10 | 0.00 | 62.06 | -0.72 | 0.00 | 0.83 |
| WAL | 1 | 81.28 | 2.71 | 0.09 | 0.02 | 69.60 | -0.34 | 0.00 | 0.89 |
| | 2 | 66.56 | 1.05 | 0.02 | 0.08 | 62.31 | -0.13 | 0.00 | 0.93 |
| WFJ | 1 | 94.17 | 0.73 | 0.01 | 0.33 | 93.65 | -1.33 | 0.01 | 0.37 |

**Table 4: Results of a single linear regression between the corrected densities ($\rho_{corr}$) as dependent variable and wet bulb**
10 **temperature ($T_w$) as well as wind speed (U) as explanatory variables for the class median values based on all filtered data points ($n_{th}$) binned into 0.5° K classes and classes of 0.5 ms$^{-1}$, respectively. The corresponding coefficient of determination ($r^2$) and the significance level (p) are presented.**

| Station | Period # | $T_w$ | | | | U | | | |
|---|---|---|---|---|---|---|---|---|---|
| | | Intercept | $d\rho_{corr}/dT_w$ | $r^2$ | p | Intercept | $d\rho_{corr}/dT_w$ | $r^2$ | p |
| KRO | 1 | 80.07 | 4.00 | 0.65 | 0.00 | 40.35 | 19.10 | 0.35 | 0.16 |
| | 2 | 76.04 | 0.99 | 0.11 | 0.35 | 64.52 | 1.29 | 0.00 | 0.90 |
| KTA | 1 | 65.45 | 1.84 | 0.12 | 0.44 | 76.20 | -14.44 | 0.41 | 0.36 |
| | 2 | 55.34 | -0.37 | 0.02 | 0.75 | 53.41 | 3.37 | 0.53 | 0.17 |
| | 3 | 67.42 | 1.51 | 0.56 | 0.01 | 65.76 | -3.82 | 0.39 | 0.10 |
| | 4 | 70.53 | 3.75 | 0.82 | 0.00 | 46.96 | 9.41 | 0.30 | 0.26 |
| WAL | 1 | 77.40 | 2.88 | 0.47 | 0.06 | 64.95 | 1.32 | 0.02 | 0.71 |
| | 2 | 64.09 | 0.97 | 0.17 | 0.21 | 59.06 | 1.50 | 0.05 | 0.57 |
| WFJ | 1 | 92.32 | 0.71 | 0.04 | 0.53 | 93.61 | -2.91 | 0.18 | 0.23 |





**Table 5:** Comparison of observed density values ($\rho_{corr}$, [kgm⁻³]) and parameterizations applying the formulas presented in section 2. Median values (m, [kgm⁻³]) are shown together with the Pearson correlation coefficient (r) and the root mean squared error (R, [kgm⁻³]) between the respective calculations and $\rho_{corr}$. Best values of the performance measures are highlighted for each station and time period using underlined bold numbers.

| Stat | Period # | $\rho_{obs}$ m | $\rho_{HP}$ m | r | R | $\rho_D$ m | r | R | $\rho_{LC}$ m | r | R | $\rho_V$ m | r | R | $\rho_J$ m | r | R | $\rho_S$ m | r | R | $\rho_L$ m | r | R |
|---|---|---|---|---|---|---|---|---|---|---|---|---|---|---|---|---|---|---|---|---|---|---|---|
| KRO | 1 | 67 | 85 | 0.28 | 14.4 | 100 | 0.45 | 23.3 | 121 | 0.44 | 44.4 | 101 | **0.47** | 25.3 | 75 | 0.29 | **0.5** | 92 | 0.36 | 18.9 | 60 | 0.40 | 15.8 |
| | 2 | 69 | 79 | 0.18 | 8.7 | 94 | 0.18 | 18.8 | 113 | 0.19 | 38.8 | 99 | 0.13 | 25.7 | 70 | **0.20** | **1.0** | 91 | 0.10 | 19.8 | 56 | **0.20** | 14.5 |
| KTA | 1 | 61 | 82 | 0.35 | 28.7 | 98 | **0.38** | 40.1 | 118 | **0.38** | 62.3 | 98 | 0.33 | 41.7 | 76 | 0.37 | 23.0 | 88 | 0.26 | 32.2 | 63 | 0.35 | **8.9** |
| | 2 | 54 | 80 | 0.14 | 22.4 | 94 | 0.21 | 33.2 | 114 | 0.21 | 53.2 | 96 | 0.27 | 35.6 | 73 | 0.12 | 14.3 | 79 | **0.36** | 22.1 | 62 | 0.05 | **4.9** |
| | 3 | 64 | 83 | 0.21 | 22.0 | 99 | **0.25** | 32.1 | 120 | 0.24 | 53.2 | 102 | 0.19 | 34.5 | 77 | 0.24 | 14.5 | 88 | 0.09 | 20.3 | 67 | 0.06 | **4.8** |
| | 4 | 59 | 88 | 0.25 | 26.7 | 103 | **0.32** | 35.7 | 126 | 0.31 | 57.1 | 103 | **0.32** | 37.6 | 82 | 0.25 | 19.5 | 83 | 0.10 | 24.2 | 64 | 0.26 | **5.4** |
| WAL | 1 | 66 | 77 | 0.26 | 16.1 | 90 | **0.33** | 23.9 | 108 | 0.32 | 43.9 | 103 | 0.25 | 32.8 | 65 | 0.24 | **0.7** | 92 | 0.04 | 17.9 | 59 | 0.10 | 5.9 |
| | 2 | 58 | 83 | 0.08 | 24.0 | 98 | 0.14 | 31.8 | 119 | 0.13 | 52.5 | 106 | **0.15** | 45.5 | 71 | 0.06 | 6.9 | 97 | -0.09 | 28.9 | 63 | 0.08 | **1.7** |
| WFJ | 1 | 83 | 79 | 0.08 | 8.0 | 94 | **0.10** | **2.1** | 113 | **0.10** | 17.6 | 106 | 0.00 | 19.1 | 63 | 0.09 | 34.6 | 89 | 0.01 | 2.7 | 70 | -0.03 | 14.6 |