# Peer review of "Kay Helfricht1, Lea Hartl1, Roland Koch2, Christoph Marty3, Marc Olefs2"

_Hydrology and Earth System Sciences, 2017_

## Referee Comment (RC1) · Anonymous Referee #1 · 13 Nov 2017

The authors explored new snow density (NSD) calculated from the data of automatic measurement of snow height and water equivalent with 1 hour interval at four high mountain observation sites. Because the data included uncertainty resulting from measurement errors, the authors tried to exclude them using several threshold values. Then they compared their NSD with estimated NSD using 7 existing empirical parameterizations with meteorological data in previous studies. Based on these analyses, they demonstrated that most of previous empirical parameterizations overestimate NSD. In this paper, they could not propose alternative parameterization of NSD because of insufficient statistical significant. Therefore their conclusion seems to be unsatisfied, but this study represents a massive effort for quality control of NSD based on measurement data and it deserves to be published HESS, as these data will most likely continue to

be a unique resource well into the future. However, the present paper needs for several improvements before publication. Especially, several parts should be simplified because of overlapping descriptions. Below I give suggestions for improvement of the arguments in the manuscript.

<Comments>

The names of observation site are different between texts, Figures, and Table 1. Please unify their names.

P6 L5-8: Did you estimate the effect of the estimation error of air pressure on the value of Tw?

P7 L14: -13 < T >= -2.5ãĆIJC in Eq.(6) should be wrong.

P7 L16-17: the ranges of root in Eq. (8) are ambiguous. Please clarify them.

P8 L4. "high HNW values are accompanied by rather high HN". Which figure shows this result? This needs to be addressed as well.

P8 L5-L7. Fig. 7 shows only wet bulb temperature while the authors discuss the air temperature in this part. Moreover, Tw of Kuehtai seems to be higher than Weissfluhjoch in Fig.7. Please check it

P9 L19-20: Mean Tw at Weissfluhjoch is not lowest in Fig. 7. It seems that the mean Tw at Wattener Lizum is lower than Weissfluhjoch. Please check it.

P9 L24-L25. I can not agree the sentence that "A relationship between NSD and Tw is obvious for Kuhtai stain between the different periods, with higher NSD for higher Tw." Which figure shows this result? This needs to be addressed as well.

P10 L12-31: The description in this part should be moved to "Data and Methods" because they explain how to control the quality of calculated NSD. Therefore, they should be before "Results".

[Figure]

P13 L1-L2: I can not agree the sentence that "The relative low densities presented in this study are . . ..". Are there any evidence or references ? This needs to be addressed as well

---

## Referee Comment (RC2) · Anonymous Referee #2 · 16 Nov 2017

The manuscript "Obtaining sub-daily new snow density from automate measurements in high mountain regions" by Helfricht et al. presents a valuable data set for new snow densities at four Alpine sites with automatic weather stations (AWS). The authors determine new snow density from the combination of observed changes of ultrasonic transducers and obtained increases in mass measured by snow pillows. They filter the data for several possible sources of errors. The data processing and filtering is adequate and offers an interesting approach to determine new snow density values automatically. The presented results are of high interest to the scientific community and are worth publishing. However, significant changes have to be made before acceptance for publication is possible. Hence, most of my major points of criticism are related to the presentation of this manuscript, I ask for a more detailed description of

the uncertainty of the used measurements. For instance, in P10 L40 you describe the uncertainty of your method to be at +-25 kg/m3. The approach by Lehning et al. has a RMSE far below this uncertainty. So how can you argue for deviations of model parameterizations while the given uncertainty of the presented derivation of new snow density is larger? Please restructure the MS while more focusing on the named aims of the paper, include subchapters.

Other major points of criticism:

1. The manuscript neglects spatial variability in between snow depth and SWE measurements. Although the authors discuss errors arising from the two measurements, there might be (and certainly is at WFJ) a spatial distance between the point measurement of snow depth and the more spatially integrating observation above snow pillows. Schmid et al. (2014 - doi: 10.3189/2014JoG13J084) found a small scale heterogeneity in HS of at least 4% at WFJ. In SWE, they observed an uncertainty of +-5% for all available measurements. It remains questionable what the Golden Standard is, however an uncertainty of 5% may exist. For this manuscript, just relative changes are being used, which might reduce errors due to spatial variability. However, such uncertainty has to be included in the discussion of the results. Especially, since all of your validation data arise from the assumption that both, the ultrasonic transducer and the pillow, measure exactly the same occurrences.

2. Another major part preventing the manuscript from publication at the current state is the presentation of the paper. First of all, the manuscript is far too long. You certainly don't make efficient use of the journal's space in relation to the information you provide. Rewriting your manuscript can reduce the number of pages by approx. 50%. Right now you provide large amounts of redundancy and not supportive information, for instance:

P3 2nd paragraph bridging effects do not need to be introduced and explained here. Just cite a respective publication e.g. Johnson and Schaefer. 2002 – doi: 10.1002/hyp.1236

P2 2nd paragraph – here you don't need to provide a review on snow crystal growth in the atmosphere.

P4 down to L30 has to be shortened significantly

P5 L5-26 and L27-31 provide redundant information with two Tables

P6 L3-8 Please shorten and refer to Olefs et al. (2010). No need for repetition of all the details.

Are you entirely sure that you need all Figures presented in the manuscript and the supplementary? Isn't it more useful to present quantities in a Tab? Especially since you only include Kuehroint within the MS. All data from Fig. 5 and corresponding Figs in the supplement can easily be concluded in a single table using maybe the coefficient of variation as measure of distribution instead box plots.

Fig. 6 (+ similar suppl.) and Tab 3 are redundant; same for Figs. 9,10 and Tab. 5.

The Discussion section is far too long and extensive.

3. The structure of the MS is not acceptable. In results you interpret the presented data i.e. P8 for numerous times, P9 L15-30, P10 L4-10 etc.. In Discussion, you do present results: P11 L29-39 and kind of introduce the topic P12 L.16ff. I suggest combining Results and Discussion in one section.

4. The presentation of equations is inacceptable as well. Please read the guidelines provided for this journal and follow them. I will certainly reject a revised version of this MS if equations remain unreadable. Multi-letter variables are not supportive in equations and according to the guidelines "should be avoided". Even worse are variables like $SD\_HN$ with a subscript t. For preparation of a manuscript it is not adequate to copy and paste equations from scripts.

5. It appeared - at least to me - that the usage of the term "threshold" is very misleading/ wrong. In my opinion, for the first time, it is correctly used within the MS on P8 L21.

[Figure]

Please explain Fig. 2 more in detail. So far, the reader gets no idea what you are intending to present with these plots.

6. The presentation of the Figs. should be improved as well. It is inadequate to use left, middle, second from right for the description of subplots. Please use letters or similar to differentiate plots.

7. Phrase like... are obvious ... in a statistically vague manner... should not appear in a conclusion. Either quantify or describe that no statistical relations can be found. You often use imprecise wording to describe coherences.

Some more minor points which have to be revised before publication:

- WFJ is not located at the N "fringe" of the Alps and in your comparison it is actually the most southern site. As a consequence, I do not accept the argument presented on P8 L7ff, which again should be part of the Discussion instead of Results.

- P8 L26ff, this is very confusing! You observe a data reduction to only 6% remaining at WFJ and to only 5% at Kuehtai. However, WFJ has the highest filtering rate, please clarify and probably rewrite emphasizing more on the periods to facilitate understanding.

- Be CONSISTENT! Apart from the equations the whole MS appears to be not thoroughly reviewed before submission, i.e. snowpack vs snow pack, Kuehtai, Kuehroint in at least 3 different writings...

- P6 L17ff you vary "thresholds" by values below the resolution limits of the instruments. I do not consider this as a threshold nor do I think that such increments are actually useful.

- The number 4 does not have to be introduced (P5 L4)

- Snow pillows actually do not measure SWE. They weigh the overlaying mass and allow for derivation of SWE

[Figure]

- Weight cannot settle P6 L27

- P6 L19 described in Anderson. . .

- Fig 7 is referenced before Fig. 4 etc

- What is "lateral bonding" P10 L39?

- . . . filtered OUT. . . P11 L13; . . .more wind influenced stations. . . P11 L41

---

## Author Comment (AC1) · 4 Dec 2017

We would like to thank the anonymous referee for the time invested reviewing our manuscript and for the positive and constructive feedback. We will revise the manuscript with respect to the comments of the referee. We will consider in detail the conclusiveness of names and abbreviations.

P6 L5-8: Did you estimate the effect of the estimation error of air pressure on the value of Tw?

We didn't estimate the effect for the calculation of Tw. Tw is only used as a threshold to exclude mixed phase precipitation. Thus, the estimation error of air pressure does not have a direct effect on the calculation of the new snow density from observation. It

might have an effect on the statistical relation of Tw and new snow density, but it can be assumed to be rather small compared to the overall variability of new snow density. Air pressure dependency of wet-bulb temperature is generally minor and only relevant for air temperatures larger than +2°C (Olefs et. al, 2010, supplemental material section b)).

P7 L14: -13°C < T >= -2.5°C in Eq.(6) should be wrong.

Yes, will be changed to -13 < T <= 2.5°C

P7 L16-17: the ranges of root in Eq. (8) are ambiguous. Please clarify them.

Will be clarified and presentation of all equations will be revised to match journal guidelines.

P8 L4. "high HNW values are accompanied by rather high HN". Which figure shows this result? This needs to be addressed as well.

If the density is low even at high HNW, then HN must be also high to achieve low density values. This can be seen in the rather few values of more than 3 mm HNW at Kühtai and Wattener Lizum station in Fig. 2.

P8 L5-L7. Fig. 7 shows only wet bulb temperature while the authors discuss the air temperature in this part. Moreover, Tw of Kuehtai seems to be higher than Weissfluhjoch in Fig.7. Please check it

Will be corrected to wet bulb temperatures at snowfall.

P9 L19-20: Mean Tw at Weissfluhjoch is not lowest in Fig. 7. It seems that the mean Tw at Wattener Lizum is lower than Weissfluhjoch. Please check it.

Wattener Lizum is lower than Weissfluhjoch . Sentence will be revised.

P9 L24-L25. I can not agree the sentence that "A relationship between NSD and Tw is obvious for Kuhtai stain between the different periods, with higher NSD for higher Tw."

Which figure shows this result? This needs to be addressed as well.

This result is based on Fig. 8, where the statistical distributions of NSD and TW are shown. This will be addressed and clarified in the revised manuscript.

P10 L12-31: The description in this part should be moved to "Data and Methods" because they explain how to control the quality of calculated NSD. Therefore, they should be before "Results".

The general structure of the manuscript will be revised taking into account this comment and the comments made by reviewer 2.

P13 L1-L2: I can not agree the sentence that "The relative low densities presented in this study are..". Are there any evidence or references ? This needs to be addressed as well

This sentence is a general assumption based on the relationship between snow density and the potential of drifting snow. Corresponding literature will be added.

Literature: Olefs, M., Fischer, A., and Lang, J.: Boundary conditions for artificial snow production in the Austrian Alps. J. Appl. Meteorol. Climatol., 49(6): 1096-113, doi: http://dx.doi.org/10.1175/2010JAMC2251.1, 2010.

---

## Author Comment (AC2) · 4 Dec 2017

We would like to thank the anonymous referee for the time invested in reviewing our manuscript, and for the positive and constructive feedback. We will revise the manuscript with special focus on the conclusiveness of names and abbreviations and rephrase the sections mentioned by the referee. The presentation and structure of the manuscript will be revised substantially.

We kindly forward the decision regarding the inclusion of all additional figures in the supplement to the editor.

1. The manuscript neglects spatial variability in between snow depth and SWE measurements. Although the authors discuss errors arising from the two measurements,

there might be (and certainly is at WFJ) a spatial distance between the point measurement of snow depth and the more spatially integrating observation above snow pillows. Schmid et al. (2014 - doi: 10.3189/2014JoG13J084) found a small scale heterogeneity in HS of at least 4% at WFJ. In SWE, they observed an uncertainty of +-5% for all available measurements. It remains questionable what the Golden Standard is, however an uncertainty of 5% may exist. For this manuscript, just relative changes are being used, which might reduce errors due to spatial variability. However, such uncertainty has to be included in the discussion of the results. Especially, since all of your validation data arise from the assumption that both, the ultrasonic transducer and the pillow, measure exactly the same occurrences.

Uncertainties from observations and snow cover variability will be addressed in more detail in a revised version of the manuscript.

2. Another major part preventing the manuscript from publication at the current state is the presentation of the paper. First of all, the manuscript is far too long. You certainly don't make efficient use of the journal's space in relation to the information you provide. Rewriting your manuscript can reduce the number of pages by approx. 50%. Right now you provide large amounts of redundancy and not supportive information, for instance: P3 2nd paragraph bridging effects do not need to be introduced and explained here. Just cite a respective publication e.g. Johnson and Schaefer. 2002 – doi: 10.1002/hyp.1236

The manuscript will be shortened and redundant information will be deleted.

P3 2nd paragraph bridging effects do not need to be introduced and explained here. Just cite a respective publication e.g. Johnson and Schaefer. 2002 – doi:10.1002/hyp.1236

Will be revised.

P2 2nd paragraph – here you don't need to provide a review on snow crystal growth in

the atmosphere.

Will be revised.

P4 down to L30 has to be shortened significantly

Will be considered.

P5 L5-26 and L27-31 provide redundant information with two Tables

Will be shortened.

P6 L3-8 Please shorten and refer to Olefs et al. (2010). No need for repetition of all the details.

Will be shortened.

Are you entirely sure that you need all Figures presented in the manuscript and the supplementary? Isn't it more useful to present quantities in a Tab? Especially since you only include Kuehroint within the MS. All data from Fig. 5 and corresponding Figs in the supplement can easily be concluded in a single table using maybe the coefficient of variation as measure of distribution instead box plots.

From our point of view, the additional figures in the supplementary materials complete the in-depth presentation of the analysis. The Kuehroint figure was included as an example, with a reference to the additional figures for interested readers. It also highlights the soundness of the data filtering and the unequal distribution of NSD, seeing as not only mean and standard deviation but also median and quartiles are been presented. We defer to the editor on the decision whether or not to include the additional figures.

Fig. 6 (+ similar suppl.) and Tab 3 are redundant; same for Figs. 9,10 and Tab. 5.

We think that not all numbers presented in Tab. 3 and Tab. 5 can be estimated from the corresponding figures, in particular the correlation coefficient and the coefficient of determination. Conversely, getting a general overview from the tables only is difficult.

The authors are in favour of presenting results in a way that appeals to readers who prefer visual information as well as those who prefer data tables and would prefer to include both figures and tables. However, we will reconsider the presentation of the data mentioned by the reviewer and aim for a more concise solution.

The Discussion section is far too long and extensive.

Will be shortened.

3. The structure of the MS is not acceptable. In results you interpret the presented data i.e. P8 for numerous times, P9 L15-30, P10 L4-10 etc.. In Discussion, you do present results: P11 L29-39 and kind of introduce the topic P12 L.16ff. I suggest combining Results and Discussion in one section.

The general structure of the manuscript will be revised substantially.

4. The presentation of equations is inacceptable as well. Please read the guidelines provided for this journal and follow them. I will certainly reject a revised version of this MS if equations remain unreadable. Multi-letter variables are not supportive in equations and according to the guidelines "should be avoided". Even worse are variables like SD_HN with a subscript t. For preparation of a manuscript it is not adequate to copy and paste equations from scripts.

Our apologies for not considering equation guidelines in the discussion manuscript. This will be revised in the next version.

5. It appeared - at least to me - that the usage of the term "threshold" is very misleading/ wrong. In my opinion, for the first time, it is correctly used within the MS on P8 L21.

We will reconsider the use of the term "threshold".

Please explain Fig. 2 more in detail. So far, the reader gets no idea what you are intending to present with these plots.

The presentation of Fig. 2 will be clarified.

6. The presentation of the Figs. should be improved as well. It is inadequate to use left, middle, second from right for the description of subplots. Please use letters or similar to differentiate plots.

Will be revised.

7. Phrase like. . . are obvious . . . in a statistically vague manner. . . should not appear in a conclusion. Either quantify or describe that no statistical relations can be found. You often use imprecise wording to describe coherences.

Will be revised.

Some more minor points which have to be revised before publication: - WFJ is not located at the N "fringe" of the Alps and in your comparison it is actually the most southern site. As a consequence, I do not accept the argument presented on P8 L7ff, which again should be part of the Discussion instead of Results.

Will be clarified.

- P8 L26ff, this is very confusing! You observe a data reduction to only 6% remaining at WFJ and to only 5% at Kuehtai. However, WFJ has the highest filtering rate, please clarify and probably rewrite emphasizing more on the periods to facilitate understanding.

Will be clarified.

- Be CONSISTENT! Apart from the equations the whole MS appears to be not thoroughly reviewed before submission, i.e. snowpack vs snow pack, Kuehtai, Kuehroint in at least 3 different writings. . .

The manuscript will be revised with focus on conclusiveness of names and abbreviations.

- P6 L17ff you vary "thresholds" by values below the resolution limits of the instruments. I do not consider this as a threshold nor do I think that such increments are actually

useful.

Even when the resolution of the instruments is of a similar magnitude, the measured values have a higher resolution of at least one order of magnitude. Fig. 2 is useful to present the influence of choosing reasonable lower limits / thresholds.

- The number 4 does not have to be introduced (P5 L4)

Will be deleted.

- Snow pillows actually do not measure SWE. They weigh the overlaying mass and allow for derivation of SWE

Will be rephrased.

- Weight cannot settle P6 L27

Will be rephrased.

- P6 L19 described in Anderson. . .

Will be revised.

- Fig 7 is referenced before Fig. 4 etc

Will be revised.

- What is "lateral bonding" P10 L39?

This is related to bridging effects and will be rephrased.

- . . . filtered OUT. . . P11 L13; . . .more wind influenced stations. . . P11 L41
Interactive

Will be revised.

---

## Author Response (AR1)

*Point-by-point reply to the comments by Anonymous Referee #1*

We would like to thank the anonymous referee for the time invested reviewing our manuscript and for the positive and constructive feedback.

The manuscript was revised with special focus on the conclusiveness of names and abbreviations.

P6 L5-8: Did you estimate the effect of the estimation error of air pressure on the value of Tw?

We didn't estimate the effect for the calculation of Tw. Tw is only used as a threshold to exclude mixed phase precipitation. Thus, the estimation error of air pressure does not have a direct effect on the calculation of the new snow density from observation.

We added the following sentence in the text: "Air pressure dependency of wet-bulb temperature is generally minor and only relevant for air temperatures larger than +2°C.", and refer to the study of Olefs et al. 2010)

P7 L14: -13°C < T >= -2.5°C in Eq.(6) should be wrong.

Corrected to -13 < T <= 2.5°C

P7 L16-17: the ranges of root in Eq. (8) are ambiguous. Please clarify them.

Presentation of the equations was revised.

P8 L4. "high HNW values are accompanied by rather high HN". Which figure shows this result? This needs to be addressed as well.

Sentence changed to: At Kühtai and Wattener Lizum station, high HNW values of more than 3 mm HNW are accompanied by rather high HN (Fig. 2).

P8 L5-L7. Fig. 7 shows only wet bulb temperature while the authors discuss the air temperature in this part. Moreover, Tw of Kuehtai seems to be higher than Weissfluhjoch in Fig.7. Please check it

Sentence was deleted.

P9 L19-20: Mean Tw at Weissfluhjoch is not lowest in Fig. 7. It seems that the mean Tw at Wattener Lizum is lower than Weissfluhjoch. Please check it.

This incorrect result was deleted.

P9 L24-L25. I can not agree the sentence that "A relationship between NSD and Tw is obvious for Kuhtai stain between the different periods, with higher NSD for higher Tw." Which figure shows this result? This needs to be addressed as well.

Sentence changed to: At Kühtai station,  median NSD and median Tw of the different periods show a relationship, with higher NSD for higher Tw  (Fig. 8, Tab. 2).

P10 L12-31: The description in this part should be moved to "Data and Methods" because they explain how to control the quality of calculated NSD. Therefore, they should be before "Results".

The general structure of the manuscript was revised and this part was moved to the "Data and Methods" section.

P13 L1-L2: I can not agree the sentence that "The relative low densities presented in this study are..". Are there any evidence or references ? This needs to be addressed as well

We decided to remove this sentence according to the more general suggestion of shortening the manuscript.

Literature:

Olefs, M., Fischer, A., and Lang, J.: Boundary conditions for artificial snow production in the Austrian Alps. J. Appl. Meteorol. Climatol., 49(6): 1096-113, doi: http://dx.doi.org/10.1175/2010JAMC2251.1, 2010.

*Point-by-point reply to the comments by Anonymous Referee #2*

We would like to thank the anonymous referee for the time invested in reviewing our manuscript, and for the positive and constructive feedback.

The manuscript was revised with special focus on the conclusiveness of names and abbreviations. The overall presentation and structure of the manuscript was revised as suggested by the referee.

We contacted the Editor regarding a suggestion on how to proceed on the supplementary material. We decided to keep all supplementary figures for additional information, but remove two figures from the text.

1. The manuscript neglects spatial variability in between snow depth and SWE measurements. Although the authors discuss errors arising from the two measurements, there might be (and certainly is at WFJ) a spatial distance between the point measurement of snow depth and the more spatially integrating observation above snow pillows. Schmid et al. (2014 - doi: 10.3189/2014JoG13J084) found a small scale heterogeneity in HS of at least 4% at WFJ. In SWE, they observed an uncertainty of +-5% for all available measurements. It remains questionable what the Golden Standard is, however an uncertainty of 5% may exist. For this manuscript, just relative changes are being used, which might reduce errors due to spatial variability. However, such uncertainty has to be included in the discussion of the results. Especially, since all of your validation data arise from the assumption that both, the ultrasonic transducer and the pillow, measure exactly the same occurrences.

HS is measured directly above the snow pillow at Kühtai station, Kühroint station and Wattener Lizum station (Fig. 1). We added the following text:

A source of uncertainty is the spatial offset between HS measurements and SWE measurements. HS is measured directly above the SWE measurement at Kühtai station, Kühroint station and Wattener Lizum station (Fig. 1). However, the footprint of the snow depth sensor may be smaller than the surface area of the pillow, and it is decreasing with increasing HS. A spatial variability of HS on the pillow may be caused by snow drift and differential snow settling or snow melt.

For the calculations within this study we used the changes in HS and SWE over a short time period only. Errors due to spatial variability in HS and SWE caused by spatial differences in energy consumption and snow drift between precipitation events are reduced. This is especially valid for the HS and SWE measurements at the matching sites.

The snow depth sensor and the snow pillow of Weissfluhjoch station are separated by 9 meters. Schmid et al. (2014) suggest a small-scale variability in HS of $\pm4.3$ % at the Weissfluhjoch station. Again, the error may be smaller due to using temporally limited changes of HS, but an additional uncertainty of $\pm5$ % can be assumed here.

2. Another major part preventing the manuscript from publication at the current state is the presentation of the paper. First of all, the manuscript is far too long. You certainly don't make efficient use of the journal's space in relation to the information you provide. Rewriting your manuscript can reduce the number of pages by approx. 50%. Right now you provide large amounts of redundancy and not supportive information, for instance: P3 2nd paragraph bridging effects do not

need to be introduced and explained here. Just cite a respective publication e.g. Johnson and Schaefer. 2002 – doi: 10.1002/hyp.1236

      We have shortened the manuscript substantially and removed redundant information.

P2 2nd paragraph – here you don't need to provide a review on snow crystal growth in the atmosphere.

      These sentences have been removed.

P4 down to L30 has to be shortened significantly

      The introduction has been shortened.

P5 L5-26 and L27-31 provide redundant information with two Tables

      Redundant information has been removed.

P6 L3-8 Please shorten and refer to Olefs et al. (2010). No need for repetition of all the details.

      This section was shortened.

Are you entirely sure that you need all Figures presented in the manuscript and the supplementary? Isn't it more useful to present quantities in a Tab? Especially since you only include Kuehroint within the MS. All data from Fig. 5 and corresponding Figs in the supplement can easily be concluded in a single table using maybe the coefficient of variation as measure of distribution instead box plots.

 AND

Fig. 6 (+ similar suppl.) and Tab 3 are redundant; same for Figs. 9, 10 and Tab. 5.

From our point of view, the additional figures in the supplementary material complete the presentation of the analysis. We think that not all of the data presented in e.g. Tab. 3 and Tab. 5 can be estimated from the corresponding figures, in particular the correlation coefficient and the coefficient of determination. In presenting figures we aim to appeal to readers who prefer visual information as well as to those who prefer numbers in tables. Thus, we kept all the figures in the supplement. We moved Fig. (4) and (6) to the supplement for shortening of the manuscript. We kept Fig. (5) in the manuscript as an example, because this figure shows nicely the distribution and the effect of including settling in the calculations.

The Discussion section is far too long and extensive.

      We have shortened the discussion section.

3. The structure of the MS is not acceptable. In results you interpret the presented data i.e. P8 for numerous times, P9 L15-30, P10 L4-10 etc.. In Discussion, you do present results: P11 L29-39 and kind of introduce the topic P12 L.16ff. I suggest combining Results and Discussion in one section.

      We have combined results and discussion. The uncertainty discussion was moved to the methods section.

4. The presentation of equations is inacceptable as well. Please read the guidelines provided for this journal and follow them. I will certainly reject a revised version of this MS if equations remain unreadable. Multi-letter variables are not supportive in equations and according to the guidelines "should be avoided". Even worse are variables like SD_HN with a subscript t. For preparation of a manuscript it is not adequate to copy and paste equations from scripts.

Our apologies for not considering equation guidelines in the discussion manuscript. All equations and variables have been edited following the journal guide lines.

5. It appeared - at least to me - that the usage of the term "threshold" is very misleading/ wrong. In my opinion, for the first time, it is correctly used within the MS on P8 L21.

The term threshold was removed from the manuscript. "Minimum values" is used instead.

Please explain Fig. 2 more in detail. So far, the reader gets no idea what you are intending to present with these plots.

We have rephrased the introduction of Fig. (2) to: Figure (2) presents the median new snow density ($\rho_{HN}$) data exceeding the respective minimum HN and HNW values. This presentation highlights the variability of $\rho_{HN}$ by using different constraints for the data filtering with respect to the high relative uncertainty of low HN and HNW values.

6. The presentation of the Figs. should be improved as well. It is inadequate to use left, middle, second from right for the description of subplots. Please use letters or similar to differentiate plots.

The presentation of the figures has been revised.

7. Phrase like. . . are obvious . . . in a statistically vague manner. . . should not appear in a conclusion. Either quantify or describe that no statistical relations can be found. You often use imprecise wording to describe coherences.

The text has been revised with respect to this issue.

Some more minor points which have to be revised before publication:
- WFJ is not located at the N "fringe" of the Alps and in your comparison it is actually the most southern site. As a consequence, I do not accept the argument presented on P8 L7ff, which again should be part of the Discussion instead of Results.

We agree and have removed this argument.

- P8 L26ff, this is very confusing! You observe a data reduction to only 6% remaining at WFJ and to only 5% at Kuehtai. However, WFJ has the highest filtering rate, please clarify and probably rewrite emphasizing more on the periods to facilitate understanding.

The different periods have been mentioned in the text.

- Be CONSISTENT! Apart from the equations the whole MS appears to be not thoroughly reviewed before submission, i.e. snowpack vs snow pack, Kuehtai, Kuehroint in at least 3 different writings. . .

The manuscript has been revised with focus on conclusiveness of names and abbreviations.

- P6 L17ff you vary "thresholds" by values below the resolution limits of the instruments.
I do not consider this as a threshold nor do I think that such increments are actually useful.

The accuracy of the instruments is of a similar magnitude as the minimum values. Actually, the measured values have a higher resolution of at least one order of magnitude (HS in mm, SWE in 1/10mm). Changing the increments doesn't change the results presented in Fig. (2).

- The number 4 does not have to be introduced (P5 L4)

Removed.

- Snow pillows actually do not measure SWE. They weigh the overlaying mass and allow for derivation of SWE

This issue has been rephrased.

- Weight cannot settle P6 L27

Changed to: Snow settling of the new snow layer caused by the weight of the ongoing snow accumulation is not taken into account.

- P6 L19 described in Anderson. . .

Corrected.

- Fig 7 is referenced before Fig. 4 etc

Revised

- What is "lateral bonding" P10 L39?

Changed to:  to avoid bridging effects.

- . . . filtered OUT. . . P11 L13; . . .more wind influenced stations. . . P11 L41
Interactive

Corrected.

[revised manuscript text omitted]
}$ | $\rho_{HP}$ | | | $\rho_D$ | | | $\rho_{LC}$ | | | $\rho_V$ | | | $\rho_J$ | | | $\rho_S$ | | | $\rho_L$ | | |
|---|---|---|---|---|---|---|---|---|---|---|---|---|---|---|---|---|---|---|---|---|---|---|---|
| | # | m | m | r | R | m | r | R | m | r | R | m | r | R | m | r | R | m | r | R | m | r | R |
| KRO | 1 | 67 | 85 | 0.28 | 14.4 | 100 | 0.45 | 23.3 | 121 | 0.44 | 44.4 | 101 | **0.47** | 25.3 | 75 | 0.29 | **0.5** | 92 | 0.36 | 18.9 | 60 | 0.40 | 15.8 |
| | 2 | 69 | 79 | 0.18 | 8.7 | 94 | 0.18 | 18.8 | 113 | 0.19 | 38.8 | 99 | 0.13 | 25.7 | 70 | **0.20** | **1.0** | 91 | 0.10 | 19.8 | 56 | **0.20** | 14.5 |
| KTA | 1 | 61 | 82 | 0.35 | 28.7 | 98 | **0.38** | 40.1 | 118 | **0.38** | 62.3 | 98 | 0.33 | 41.7 | 76 | 0.37 | 23.0 | 88 | 0.26 | 32.2 | 63 | 0.35 | **8.9** |
| | 2 | 54 | 80 | 0.14 | 22.4 | 94 | 0.21 | 33.2 | 114 | 0.21 | 53.2 | 96 | 0.27 | 35.6 | 73 | 0.12 | 14.3 | 79 | **0.36** | 22.1 | 62 | 0.05 | **4.9** |
| | 3 | 64 | 83 | 0.21 | 22.0 | 99 | **0.25** | 32.1 | 120 | 0.24 | 53.2 | 102 | 0.19 | 34.5 | 77 | 0.24 | 14.5 | 88 | 0.09 | 20.3 | 67 | 0.06 | **4.8** |
| | 4 | 59 | 88 | 0.25 | 26.7 | 103 | **0.32** | 35.7 | 126 | 0.31 | 57.1 | 103 | **0.32** | 37.6 | 82 | 0.25 | 19.5 | 83 | 0.10 | 24.2 | 64 | 0.26 | **5.4** |
| WAL | 1 | 66 | 77 | 0.26 | 16.1 | 90 | **0.33** | 23.9 | 108 | 0.32 | 43.9 | 103 | 0.25 | 32.8 | 65 | 0.24 | **0.7** | 92 | 0.04 | 17.9 | 59 | 0.10 | 5.9 |
| | 2 | 58 | 83 | 0.08 | 24.0 | 98 | 0.14 | 31.8 | 119 | 0.13 | 52.5 | 106 | **0.15** | 45.5 | 71 | 0.06 | 6.9 | 97 | -0.09 | 28.9 | 63 | 0.08 | **1.7** |
| WFJ | 1 | 83 | 79 | 0.08 | 8.0 | 94 | **0.10** | **2.1** | 113 | **0.10** | 17.6 | 106 | 0.00 | 19.1 | 63 | 0.09 | 34.6 | 89 | 0.01 | 2.7 | 70 | -0.03 | 14.6 |

---

## Author Response (AR2)

*Point-by-point reply to the comments by Anonymous Referee #2*

We would like to thank the anonymous referee for the time invested in reviewing our manuscript, and for the positive and constructive feedback.

- I still consider that subchapters especially for Results and Discussion will improve understanding and readability significantly. At the current stage, you lack a clear structure for this section. For instance results for the parameterizations are already discussed at P7 L24ff and all after P9 L18. However, at P7 L32, you discuss data filtering and minimum requirements. This is all very confusing. Most of P9 L18ff could be incorporated into a subsection parameterizations or similar. In the introduction, you state 3 main questions this work should address. So why don't you structure Results and Discussion such that you answer those questions. By the way, question #2 is not even mentioned/ answered in the conclusion.

The results and discussion section was structured by subchapters according to the 3 main questions as suggested by the referee. We also reorganized the conclusions and answered question #2.

- I still consider redundancies as major problem. You even name the redundancies in your manuscript: e.g. P9 L29-30 and many other occurrences - Fig. 3 AND 7 AND Table 5; same page L21: Fig. 3 AND 7 AND 8 AND Table 5. What is the purpose of Figs. 7 and 8 next to Tab. 5. I absolutely cannot detect any benefit from those 2 Figures apart from the fact that they are hard to read.

Same for supplemental Figs S17-S24. You reference those Figs once in the whole manuscript, however they do not provide any additional information. The fact that regressions are poorly constrained/ not present is documented by Tables 3 and 4. In addition, you spend a significant part of the discussion section on the topic of linear regression between parameters (without mentioning such issue as major point of your manuscript). The presentation of 2 tables and EIGHT additional figures in the supplement - apart from the fact that the references in P8 L17 are wrong – Is certainly not beneficial. Again, I highly recommend to shorten!

We shortened the manuscript and removed redundancies by:

- Removing the former Fig. 3, because the results are also represented by the Fig. 7 and 8
- Removing Table 3 and presenting former Table 4 only.
- Removing the Fig. S17 – S24 from the supplemental.

We kept Fig. 7/8 and Table 5 and argue that the numbers given in the table are not provided in total by the figures. In turn, the figures give a better overview.

- Consistency is still an issue. Stations still have various labels in the supplementary material!

We revised the manuscript thoroughly to fulfil consistency.

P1 L21, L22 please quantify in the abstract: The terms "partly visible", "crucial parameter" don't provide any information to the reader.

Sentences have been changed to: Applying simple linear regressions between new snow density and wet bulb temperature based on the measurements data resulted in significant relationships ($r^2 > 0.5$ and $p \leq 0.05$) for single periods at individual stations only. Higher new snow density was calculated for the highest elevated and most wind exposed station location.

P1 L25 for better investigations...? for more reliable investigations...?

Changed to: more reliable.

P2 L15 it is recommended to provide either the first reference for some statement or the most relevant (Gubler 1981 is probably more suitable as reference here doi:10.3189/S002214300001131X)

We thank for this advice and added Gubler (1981) to the reference.

P7 L27 better: ...parameterizations overestimate \rho_{HN} in comparison with values computed....

Sentence was revised.

P7 L28-29 Please rephrase this statement. It is really hard to get what you would like to express here.

This sentence corresponding to the former Fig. 3 was deleted.

P8 L4ff Fig. 4 just presents data distribution for station KRO!

Sentence revised to: Figure 4 shows the distribution of the filtered values at Kühroint station as a representative for all stations and periods (Fig. S10 to S17 in the supplement).

P8 L6 where do you show the hourly resolution? ... change substantially from one hour to the next... This is not documented in this Fig, at least for my understanding.

Some of the dots are representing events over several continuous hours. Since we agree with the referee that this cannot easily be read from the figure, we deleted this clause.

P8 L8-11 rephrase impossible to read!

Sentence rephrased. The correction of the HN underestimation caused by settling of the snowpack during snowfall leads to an average reduction of mean $\rho_{HN}$ of 10.2 kg m$^{-3}$ with a standard deviation ($\sigma$) of 2.6 kg m$^{-3}$. This corresponds to 13.5 % with a $\sigma$ of 3.7 % (Table 2).

P8 L32ff remove the statement "Therefore,..." This is not a goal of the study so no need for describing that something hasn't been performed.

The statement was removed.

P8 L37 wind speeds WERE observed... at WFJ station. The remaining part of the sentence is just another redundant phrase.

The sentence was corrected and shortened.

P8 L41 \rho_{HN} remove the point and insert space!

Corrected.

P9 L1 Snow grains are fragmented...

Revised.

P9 L41 Space after brackets

Revised.

P9 L41 Essentially, ....fundamental relation between snow density and TA. This statement contradicts all your regression data and previous statements on regressions!

We revised this sentence. Nevertheless, the coefficients of determination from Table 3 correspond well to the square of the Pearson correlation values presented in Table 4.

[revised manuscript text omitted]
}$ | $\rho_{HP}$ | | | $\rho_{D}$ | | | $\rho_{LC}$ | | | $\rho_{V}$ | | | $\rho_{J}$ | | | $\rho_{S}$ | | | $\rho_{L}$ | | |
|------|--------|-----|-----|------|------|-----|------|------|-----|------|------|-----|------|------|-----|------|------|-----|------|------|-----|------|------|
| | # | m | m | r | R | m | r | R | m | r | R | m | r | R | m | r | R | m | r | R | m | r | R |
| KRO | 1 | 67 | 85 | 0.28 | 14.4 | 100 | 0.45 | 23.3 | 121 | 0.44 | 44.4 | 101 | **0.47** | 25.3 | 75 | 0.29 | **0.5** | 92 | 0.36 | 18.9 | 60 | 0.40 | 15.8 |
| | 2 | 69 | 79 | 0.18 | 8.7 | 94 | 0.18 | 18.8 | 113 | 0.19 | 38.8 | 99 | 0.13 | 25.7 | 70 | **0.20** | **1.0** | 91 | 0.10 | 19.8 | 56 | **0.20** | 14.5 |
| KTA | 1 | 61 | 82 | 0.35 | 28.7 | 98 | **0.38** | 40.1 | 118 | **0.38** | 62.3 | 98 | 0.33 | 41.7 | 76 | 0.37 | 23.0 | 88 | 0.26 | 32.2 | 63 | 0.35 | **8.9** |
| | 2 | 54 | 80 | 0.14 | 22.4 | 94 | 0.21 | 33.2 | 114 | 0.21 | 53.2 | 96 | 0.27 | 35.6 | 73 | 0.12 | 14.3 | 79 | **0.36** | 22.1 | 62 | 0.05 | **4.9** |
| | 3 | 64 | 83 | 0.21 | 22.0 | 99 | **0.25** | 32.1 | 120 | 0.24 | 53.2 | 102 | 0.19 | 34.5 | 77 | 0.24 | 14.5 | 88 | 0.09 | 20.3 | 67 | 0.06 | **4.8** |
| | 4 | 59 | 88 | 0.25 | 26.7 | 103 | **0.32** | 35.7 | 126 | 0.31 | 57.1 | 103 | **0.32** | 37.6 | 82 | 0.25 | 19.5 | 83 | 0.10 | 24.2 | 64 | 0.26 | **5.4** |
| WAL | 1 | 66 | 77 | 0.26 | 16.1 | 90 | **0.33** | 23.9 | 108 | 0.32 | 43.9 | 103 | 0.25 | 32.8 | 65 | 0.24 | **0.7** | 92 | 0.04 | 17.9 | 59 | 0.10 | 5.9 |
| | 2 | 58 | 83 | 0.08 | 24.0 | 98 | 0.14 | 31.8 | 119 | 0.13 | 52.5 | 106 | **0.15** | 45.5 | 71 | 0.06 | 6.9 | 97 | -0.09 | 28.9 | 63 | 0.08 | **1.7** |
| WFJ | 1 | 83 | 79 | 0.08 | 8.0 | 94 | **0.10** | **2.1** | 113 | **0.10** | 17.6 | 106 | 0.00 | 19.1 | 63 | 0.09 | 34.6 | 89 | 0.01 | 2.7 | 70 | -0.03 | 14.6 |